# A case study of wind farm effects using two wake parameterizations in WRF (V3.7.1) in the presence of low level jets

Xiaoli G. Larsén[1] and Jana Fischereit[1]

[1]Wind Energy Department, Technical University of Denmark, Frederiksborgvej 399, Building 125, 4000 Roskilde, Denmark

**Correspondence:** Xiaoli G. Larsén (xgal@dtu.dk)

**Abstract.** While the wind farm parameterization by Fitch et al. (2012) in Weather Research and Forecasting (WRF) model has been used and evaluated frequently, the Explicit Wake Parameterization (EWP) by Volker et al. (2015) is less well explored. The openly available high frequency flight measurements from Bärfuss et al. (2019) provide an opportunity to directly compare the simulation results from the EWP and Fitch scheme with in situ measurements. In doing so, this study aims to compliment the recent study by Siedersleben et al. (2020) by (1) comparing the EWP and Fitch schemes in terms of turbulent kinetic energy (TKE) and velocity deficit, together with FINO 1 measurements and Synthetic Aperture Radar (SAR) data and (2) exploring the interactions of the wind farm with Low Level Jets. This is done using a bug-fixed WRF version that includes the correct TKE advection, following Archer et al. (2020).

Both the Fitch and the EWP schemes can capture the mean wind field in the presence of the wind farm consistently and well. TKE in the EWP scheme is significantly underestimated, suggesting that an explicit turbine-induced TKE source should be included in addition to the implicit source from shear. The value of the correction factor for turbine-induced TKE generation in the Fitch scheme has a significant impact on the simulation results. The position of the LLJ nose and the shear beneath the jet nose are modified by the presence of wind farms.

## 1 Introduction

Offshore wind energy has been developing fast in recent years. Consequently, wind farms are growing bigger and bigger in both capacity and spatial sizes. For instance in the North Sea, a farm can extend over tens of kilometers, and merge with neighbouring farms, resulting in a cluster size of several thousands of km$^2$, e.g. the Hornsea area (7240 km$^2$); see 4Coffshore and Díaz and Guedes Soares (2020) for an overview of the current status of offshore wind farms. Wind turbines and farms extract momentum from the atmospheric flow and interact with it, causing reduction in wind speed and changes in turbulence in the wake regions. To assess such impact over areas with sizes of modern farm-clusters, mesoscale modeling has shown to be a useful tool in including synoptical and mesoscale wind variability. Several mesoscale models have been used to study the wind farm effects, and the Weather Research and Forecasting (WRF) model (Skamarock et al., 2007) is the most-used

mesoscale model for studying this subject, according to a recent review by Fischereit et al. (2021). There are mainly two kinds of methods to parameterize the effects of wind farms on the atmosphere: one is the implicit method that parameterizes the effects through an increase in surface roughness length, and the other is the explicit method that parameterizes the effects through an elevated momentum sink. In connection with the use of WRF, the wind farm parameterization (WFP) scheme (Fitch et al., 2012), called the Fitch scheme here, and the explicit wake parameterization (Volker et al., 2015), called the EWP scheme here, are the two most commonly applied explicit wind farm parameterizations. Most previous studies used the Fitch scheme (Fischereit et al., 2021).

The Fitch scheme has long been implemented in WRF, which makes it convenient for users, regarding further development, investigation, application and validation. The EWP scheme, on the other hand, is not included in the official WRF repository and it has not been explored and validated as frequently. Studies comparing the two schemes on the calculation of the turbulent kinetic energy (TKE) are thus limited. Volker et al. (2015), Volker et al. (2017), Catton (2020), Pryor et al. (2020) and Shepherd et al. (2020) are the few, with the first three addressing offshore wind farms, and the last two onshore wind farms. These studies consistently show that the Fitch scheme generates significantly larger wind-farm-induced TKE values than the EWP scheme does. The two schemes differ with respect to their treatment of turbine-induced forces in the momentum equation as well as to their treatment of wind farms as a source of TKE. In the Fitch scheme, the turbine-induced force is represented by a local thrust force (as a function of the thrust coefficient) acting on the turbine-swept area. In the EWP scheme, a grid cell averaged drag force is applied that accounts for a sub-grid scale vertical wake expansion based on the concept from Tennekes and Lumley (1972). With respect to TKE, in the Fitch scheme wind turbines are treated as an explicit source of TKE. By neglecting mechanical losses, turbine-induced TKE is a function of the difference between the power and the thrust coefficients. While in the EWP scheme, no explicit source term is considered for wind farm related TKE and the turbine-induced TKE arises solely from the shear production in the wind farm wake.

TKE describes the fluctuation of kinetic energy and it is related to the turbulence, which is a key wind-energy application parameter. The modelling of wind-farm-induced TKE from the Fitch scheme has been previously evaluated in a number of case studies with measurements from profiling lidars (Lee and Lundquist, 2017a, b). With considerable uncertainties embedded with the lidar technique, Lee et al. showed that TKE from the Fitch scheme can capture the general pattern from the measurements. Siedersleben et al. (2020) (hereinafter S2020) used in situ high-frequency airborne measurements to evaluate TKE from the Fitch scheme in WRF for three case studies. They found that the Fitch scheme overestimates the TKE on the upwind side of the wind farm and underestimates it on the downwind side. They also noted that capturing the background meteorology is crucial to evaluate the performance of the scheme, which they managed only within their case study II.

During the case study II of S2020, Low Level Jets were present over the area. LLJs over the Southern North Sea are mostly associated with relatively warm continental air being advected over cooler sea surface, where a stable internal boundary layer develops, causing quasi-frictional decoupling and an acceleration of air mass. The phenomena are rather common in coastal regions and they have been studied in a long list of literature, (e.g. Smedman et al., 1993, 1995; Dörenkämper et al., 2015; Wagner et al., 2019; Kalverla et al., 2019). Wagner et al. (2019) showed that LLJs are a common phenomenon in the Southern North Sea area: by analysing of one-and-an-half year of campaign measurements using lidar and a passive microwave

radiometer measurements over the Southern North Sea, they found that "LLJs occurred at 14.5% of the time (449 of 3107 measurements) and on 64.8% (162 of 250) of the days". Flow from the southwest (such as case II) is one of the favourable conditions in forming LLJs in this area. Wagner et al.'s data show that LLJs in association with flow from the south and the southwest typically have a jet nose (wind speed maximum) height between 200 m and 300 m. This is expected to have non-negligible impact on the turbine performance, which is not only related to the increased wind resources, but also to the unusual vertical distribution of wind shear, with enhanced shear beneath the jet nose and negative shear above it. Thus, turbulence is also affected, causing considerable additional uncertainties in the estimation of relevant key parameters such as turbulence intensity, gust and loads. It has not yet been documented in the literature how the structure of LLJ is affected by the presence of large wind farms. There also lack published studies showing wind farm wakes in the presence of LLJs.

The purpose of this study is thus two folds: First, it refers to what S2020 pointed out: "For comparison, it would be interesting to simulate case study II with the wind farm parameterization of Volker et al. (2015)". With the availability of the high frequency velocity measurements published in Bärfuss et al. (2019a), this study contributes to this knowledge gap through revisiting the case study II in S2020 and comparing the wind farm effects modelled through the EWP and the Fitch schemes. This study will thus also be the first to use measurements to verify the calculations of TKE in the EWP scheme. Second, we study "case study II" from 14 Oct 2017 in S2020 to examine the the wind characteristics under the impact of both wind farm wakes and LLJs.

The methods used here are introduced in Sect. 2, including the measurements and the WRF model setup. The results will be presented in Sect. 3, followed by discussions in Sect. 4 and conclusions in Sect. 5.

## 2   Method

The case study from 14 Oct 2017 is modeled using WRF v3.7.1 with both the EWP and Fitch wind farm parameterization schemes; details of the model setup are given in Sect. 2.2. The model output will be analyzed together with various measurements in line with S2020. These measurements are introduced in Sect. 2.1. In this study, time in both the measurements and the modeled data is presented in UTC.

### 2.1   Measurements

Case study II from S2020 took place on 14 Oct 2017. Along with warm air advection from land to sea, LLJs formed. Wind farm wakes were generated, which is obvious from the Synthetic Aperture Radar (SAR) data as the streaks of reduced wind speed, as shown in their Fig. 4a, which is re-produced here in Fig. 1a. Three types of measurements are used here to study this event, which will be described individually in the following.

The first are the publicly available airborne measurement data published in Bärfuss et al. (2019a), described in Lampert et al. (2020) and Platis et al. (2018), and analyzed in S2020. The measurements are briefly introduced here and their details can be found in these publications. The flight track on the 14 Oct 2017 is re-produced here in Fig. 2. The transect-flight over the wind farm is indicated in green and labelled by "a". The colored blocks labeled with digits 1 to 6 indicate profiling flights along the track from the surface to about 600 m. The start and end time of each profiling flight are provided in Table 1. The profile data

provide background information, and are not affected by wind farm wakes (see Fig. 2 for their position relative to the farms). We call these data "profile-flight". In contrast, the transect flight above the wind farm at 250 m height is affected by the farm. We call these data "transect-flight".

We downloaded the flight data from Bärfuss et al. (2019a). The data include, among others, temperature $T$ (K), pressure $P$ (hpa), and the along-wind, cross-wind and vertical-wind components, $u$, $v$ and $w$, respectively. Following S2020, we calculated the potential temperature $\theta$ from temperature $T$ and pressure $P$ using $\theta = T \cdot (P_0/P)^{0.2859}$, where $P_0 = 1000$ hPa. The flight measurements are sampled at a frequency of 100 Hz at an aircraft ground speed of 66 ms$^{-1}$, corresponding to a horizontal resolution of 0.66 m (Platis et al., 2018). The flights over Godewind 1 were conducted at an elevation of 250 m, slightly above the rotor top (187 m, with hub height 110 m and diameter 154 m (S2020), see also Table 4). For the analyses of the vertical profile, we averaged the profile-flight data over a vertical interval of 10 m. For the transect-flight data, TKE is calculated following $TKE = 0.5 \cdot (\sigma_u^2 + \sigma_v^2 + \sigma_w^2)$, where $\sigma_u$, $\sigma_v$ and $\sigma_w$ are standard deviation of the three wind components. The standard deviations are derived over data length of both 2 km and 1.5 km. The choice of the data length is made following the argument in Platis et al. (2018) for the turbulence length scales. They used a data length of 1.5 km. Given the background wind speed approximately 10 - 15 ms$^{-1}$, the time scales over both 2 km and 1.5 km are on the order of a couple of minutes, which is a reasonable integral time scale for separating boundary-layer turbulence and external fluctuations. Our analyses were made using both 2 km and 1.5 km, but are presented only for 2 km, in order to match the spatial horizontal resolution of the WRF model; details are given in Sect. 3.

Wind farms included in the WRF modeling are shown on a larger map in Fig. 3a, and the details of these farms are provided in Table 4. Figure 3b is a close-up of Fig. 3a over the marked area, with the two closest consecutive rows of WRF model grids shown over the Godewind 1 farm (in black and red), covering the flight track-a (in green). An additional row of the WRF grid points (in purple) is chosen east of the farm in the wake affected area. These three rows are denoted as transect-black, transect-red and transect-purple according to the colors in our analysis. They will be used for analyzing the transect distribution of wind speed and TKE.

The second dataset originates from the FINO 1 met mast. In Fig. 3b, the location of the FINO 1 mast is marked with $F1$. Note that for our studied case, with a wind direction from about 240° (Fig. 5), the flow passes the wind farm Borkum Riffgrund before reaching $F1$, resulting in reduced wind speed downwind of the farm, including at FINO 1 (see Fig. 1a). The 10-min values of wind speed from 30 m to 100 m, and wind direction from 30 m to 90 m on 14 Oct are from FINO 1 utilized in this study.

The third measurement type is the SAR data. The wind farm wakes can be seen as reduced wind speed in Fig. 1a, where the wind field was retrieved from ENVISAT SAR at 17:17 UTC on 14 Oct 2017. The retrieval uses the empirical relationship between the 10-m wind speed and the radar backscatter that depends on the local wind-generated wind waves (Valenzuela, 1978; Hersbach et al., 2007). The spatial resolution of the SAR data shown in Fig. 1a is about 500 m. Fig. 1a is made from more than one SAR scenes; even though the farm wake pattern is continuous across scenes, there seems to be an artificial change in wind speed east of about 6.5°E, which was also present in S2020. Due to these uncertainties, the SAR data will only be analyzed qualitatively.

## 2.2 Modeling

Important elements for accurately simulating LLJs using WRF include model domain configuration, initialization and boundary forcing data, horizontal and vertical spatial resolutions, PBL schemes etc., as explored in e.g. Nunalee and Basu (2014); Wagner et al. (2019); Kalverla et al. (2019); Siedersleben et al. (2020); Tay et al. (2020). These studies suggested different best choices for the WRF setup in order to capture the LLJ characteristics. In our setup, we followed the general recommendations in literature. These studies recommend a horizontal spatial resolution of 2 km in the innermost model domain, and a large number of vertical model levels, e.g. 80 with 21 in the lowest 200 m. They also suggest that ERA 5 data (ERA5) is good as the initial and boundary forcing. Some studies show the MYNN Planetary Boundary Layer (PBL) scheme outperforms other schemes (Tay et al., 2020) and some other studies suggest that MYNN PBL performs fine but not as good as QNSE scheme (Tay et al., 2020; Nunalee and Basu, 2014). However, since the Fitch scheme can only be used in connection with MYNN to calculate TKE developments, we were forced to use the MYNN PBL scheme.

We use WRF version 3.7.1 to simulate this case. The model contains three nested domains (Fig. 4) and the spatial resolutions are 18, 6 and 2 km for domain I, II and III, respectively. Following the suggestion by S2020 and in agreement with other sensitivity studies (Tomaszewski and Lundquist, 2020; Lee and Lundquist, 2017a), we use 80 vertical layers with 21 layers below 200 m with a thickness of about 10 m. Table 2 lists the paramterization schemes regarding PBL scheme, microphysics, radiation, land use, sea surface temperature (SST) and forcing data.

The development of LLJs has shown to be sensitive to the domain configuration. In our experiments, LLJs failed to develop when the southern land area included in the domain was too small, likely caused by an unsuccessful development of the stable internal boundary layer associated with warm air advection. This problem is solved by including more area in the model domain.

The WRF simulation starts at 6:00 on 14 Oct 2017 and runs to the end of day. The simulation captures the development of LLJs and it is sufficiently long to be compared to the available measurements. The model outputs are 10-min instantaneous values from each time step, including the longitudinal wind component $U$, the meridional wind component $V$, potential temperature $\theta$ and $QKE$, from which we further calculated the wind speed, the wind direction and TKE ($0.5 \cdot QKE$).

The simulation has been run in three modes, one with the Fitch scheme, one with the EWP scheme and one without wind farms. Table 3 shows the complete list of all simulations and how each simulation is referenced in the text.

In WRF, advection of TKE can be turned either on or off. Several studies (Tomaszewski and Lundquist, 2020; Siedersleben et al., 2020), including S2020, explored the impact of deactivated and activated TKE advection in connection with the Fitch parameterization. However, a bug in WRF versions 3.5 to 4.2.1 lead to an incorrect integration of the TKE generated by wind farms into the overall TKE field as reported in Archer et al. (2020). In addition to providing a fix for this bug, they also introduced a correction factor $\alpha$, so that the wind-farm-induced TKE in the Fitch scheme is now calculated as $C_{TKE} = \alpha(C_T - C_P)$. Thus, $\alpha$ is used to adjust the magnitude of turbine-induced TKE. Based on their comparisons of a single turbine with Large Eddy Simulations, Archer et al. (2020) recommended $\alpha$ to be set to a value smaller than 1, with 0.25 being the default value.

We integrated the bug-fix in WRF v3.7.1 as described in the zenodo-repository related to our study (Larsén and Fischereit, 2021). Using this version, we conduct three experiments using the Fitch scheme: advection turned off (denoted as Fitch-off, see Table 3), and advection turned on with $\alpha = 1$ (Fitch-on-1) and turned on with $\alpha = 0.25$ (Fitch-on-0.25). For the experiments with EWP and no wind farm, TKE advection is turned on by default, denoted as EWP and NWF, respectively. Most analysis will be based on the above-mentioned five simulations. However, in the discussion (Sect. 4) we also comment on calculations

from WRF before the bug-fix (Fitch-on-old), in order to have a rough understanding of relevant results from the literature. For the sake of completeness, the simulations with the EWP scheme and no wind farms are also done with advection turned off, and they are denoted as EWP-off and NWF-off, respectively. The results of EWP-off and NWF-off will only be discussed briefly for comparison.

To include the effects of wind farms in the simulations, information on turbine location, hub height, rotor diameter, power

coefficient and thrust coefficient are required. In our simulation, the locations of the turbines from the wind farms shown in Fig. 3 are obtained from three different sources: (1) Bundesnetzagentur (2019) for most German wind farms (2) Energistyrelsen (2020) for Danish wind farms and (3) for other wind farms not included in these two data sets turbine locations have been derived from SAR images in (Langor, 2019) and manually corrected to fit the wind farm shapes and turbine numbers from emodnet (Emodnet, 2020). For the simulations in this study only wind farms built before 2018 are included in accordance with

the measurement time (Table 4). In S2020, three types of turbine are used, with Siemens SWT 6.0-154 for Godewind 1 and 2, Siemens SWT 3.6-120 for Meerwind Süd Ost and Senvion 6.2 for OWP Nordsee Ost (see Table 3 in S2020). They used the thrust and power coefficients of Siemens SWT 3.6-120 onshore for all turbines implemented in the simulation. In our study, we used a different turbine type for each wind farm according to the sources introduced above as far as they are available to us. For Alpha Ventus or BARD Offshore, we could not obtain the thrust and power coefficients for the actual turbine, therefore

we used the power and thrust curves of M5000-116 that are scaled from the NREL 5 MW turbine. The Senvion 6.2M126 turbine in the Nordsee One, OWP Nordergründe and OWP Nordsee Ost were similarly scaled from the DTU 10 MW reference turbine. Ohter power and thrust curves have been taken from Langor (2019) or from WAsP (http://www.wasp.dk/). In Table 4 the wind farms are listed with the turbine model used in the simulations. We used an initial length scale of 1.6 for the EWP scheme related to the subgrid scale vertical wake expansion. We also conducted simulations using the length scale 1.5 and 1.7

and found that the difference is negligible. In the literature, the length scale values 1.5 and 1.7 have been used (Badger et al., 2020; Volker et al., 2017, 2015) and Volker et al. (2015) shows that the difference of using the values between 1.5 and 1.9 is negligible.

## 3  Results and Analysis

### 3.1  Low Level Jets

With warmer air moving from the land over the sea in the direction of about $240°$, a stable boundary layer (SBL) developed over the sea, as can be seen from the profile-flight data, shown in Fig. 5a and c. The modeled potential temperature $\theta$ profiles consistently suggest the presence of the SBL, though the increasing of $\theta$ with height within the lowest 300 m is slightly smaller

than the measurements. The direction veering is well captured in the lowest 300 m. Above 300 m, the measurements suggest a persistent wind direction of about 250°, while the modeled wind vector continues turning an additional $10 - 20°$. LLJs are observed during these profiling flights, as shown in the distribution of wind speed with height in Fig. 5g; note that the time and location of these flights are different (Table 1 and Fig. 2). Fig. 5g clearly suggests that the wind structure of the LLJs is highly variant over time and space. This is also true for the corresponding modeled LLJs, see Fig. 5g. Several of the measured wind speed profiles have more than one jet nose, with the lowest ones beneath 200 m and the highest ones at $350 - 400$ m. This suggests a variation of the internal boundary layer in time and space, in associated with the flow from the land. The model captures the jets at the level $200 - 400$ m. At the same time, both measured and modeled TKE decrease generally with height, with the modeled values being larger, see Fig. 5e and f. While the mean TKE values from the measurements are relatively small, their fluctuations are two times larger. These are not shown here in order to avoid too much noise in the plot. As none of these profile-flight data are affected by wind farm wakes, the modeled data are the same for NWF, the Fitch and the EWP schemes. Therefore, in Fig. 5, only results from EWP are shown.

With the wind from the southwest direction, at the FINO 1 site, the LLJ structure is affected by the wake effect originating from the Borkum Riffgrund wind farm. Figure 6 shows the wind speed profiles at FINO 1 during three 1-h periods, with two during the flight periods (Fig. 6a and b) and one later in the afternoon (Fig. 6c). In each figure we show in addition to the measurements (OBS), five simulations from NWF, EWP, Fitch-on-1, Fitch-on-0.25 and Fitch-off. Each profile is a 10-min value, for both measurements and model data. The measurements only reach up to 100 m, which is way beneath the jet noses. From Fig. 6, it can be observed that the Fitch scheme in general results in smaller wind reduction below hub height than the EWP scheme, but larger wind speed reduction above hub height. Thus the average values from the surface to the rotor top height are comparable between the two schemes in this situation. For these wind profiles, Fitch-on-1 and Fitch-off are rather similar, whereas Fitch-on-0.25 shows a distinct kink at rotor top height, corresponding to considerably larger wind reduction during 14:00 to 16:00. The kink is present in all three Fitch simulations, but it is absent in EWP. From Fig. 6, it is also clear that the wind speed reduction is not limited to the rotor area, but up to the jet nose. Overall, the EWP scheme provides larger shear and better values at measurement heights for Fig. 6a and b when the jet nose is high, but the Fitch scheme describes better the profiles at the measurement heights for Fig. 6c when the jet nose is low. It is also worth noting that the position of the LLJ nose is higher in the presence of the farm wake effect according to the WRF modeling.

These characteristics are also examined for point A as shown in Fig. 3. Point A is inside the Godewind 1 wind farm area and part of the flight leg, transect-a. Data at 15:00 is used, which is in between the data used for Fig. 6a,b and close to flight number four (Table 1). Here, both the wind speed profiles and the TKE profiles are compared between the five simulations in Fig. 7a and Fig. 7b, respectively. The above descriptions of the wind speed for FINO 1 are also true for point A, as can be seen in Fig. 7a. In the absence of wind farms (NWF), TKE decreases with height, as in Fig. 5. In the presence of farm wakes, both for EWP and Fitch, TKE values increase with height up to the rotor top and then decreases again to a value matching the free stream value. With the TKE advection turned on, TKE at point A is smaller than without advection. TKE in Fitch-on-1 is only slightly smaller than TKE in Fitch-off, whereas TKE in Fitch-on-0.25 is considerably smaller due to the smaller TKE source.

This figure also shows that, over the rotor area, the variation of TKE with height from the EWP scheme is smoother and the magnitude of TKE significantly smaller in comparison with the Fitch scheme (Fig. 7b).

## 3.2 Wind farm wake effects

The vertical profile of wind speed at the FINO 1 site (Fig. 6) clearly shows the wake effect from the upstream Borkum Riffgrund wind farm. Figure 8 compares the measured and modeled time series of wind speed and wind direction at FINO 1. Here the modeled values at FINO 1 are weighted between two closest grid points (one inside and one outside the farm) according to the distances between the grid points and the mast location. This is done because the closest grid point to FINO 1 is inside the farm, while in reality, FINO 1 is on the edge, and outside of the farm. The three model modes (NWF, Fitch, EWP) provide the same wind direction calculations at 90 m which follow the measurements well until late in the afternoon when the modeled winds are more westerly than in reality (Fig. 8b). The wake impact on the wind speed at 100 m is clear, as shown in Fig. 8a. Between 12:00 and 24:00 on 14 Oct, the difference between the measured and modeled mean wind speed, $\langle \Delta U \rangle$, the standard deviation of the difference $STD$ and the absolute difference $\langle |\Delta U| \rangle$ are shown in Table 5 for the five simulations. By subtracting $\langle \Delta U \rangle$ of EWP from that of NWF, it suggests that, during this period, the wake effect is on average about 1.2 m s$^{-1}$ when calculated using the EWP scheme. Similarly, it is 1.7 m s$^{-1}$ when calculated using Fitch-on-1. Fitch-on-0.25 gives rather significantly larger wind reduction, which is about 2 m s$^{-1}$. Without taking the wind farm wakes into account, WRF overestimates the wind speed at 100 m by 1.41 m s$^{-1}$. Overall, the Fitch scheme slightly overestimates and the EWP scheme slightly underestimates $\langle \Delta U \rangle$.

The wind farm wakes are visible from the SAR image at 17:17 UTC from Fig. 1a. The corresponding 10-m wind speed from WRF using Fitch-on-0.25, Fitch-on-1 and EWP are shown in Fig. 1b, c and d, respectively. Even though WRF misses detailed patterns as in the SAR image (e.g. streaks and waves), the farm wake patterns are consistent. Comparing the wind speeds in the wake shadows and the surrounding free stream, both SAR data and WRF output suggest a wind reduction of about $1.5 - 2$ m s$^{-1}$, with the wind speeds in the farm wakes in the range of $7 - 8$ m s$^{-1}$, and the free stream wind speeds in the range of $9 - 10$ m s$^{-1}$. To make the wind farm wake effect more visible, the difference of the wind speed at 10 m between EWP and NWF, between Fitch-off and NWF-off, between Fitch-on-1 and NWF and between between Fitch-on-0.25 and NWF are shown in Fig. 9. Here one can see that the wake-caused wind speed reduction at 10 m is about 2.5 ms$^{-1}$ inside the farm, and a reduction of 0.5 ms$^{-1}$ can extend between ten and a hundred of kilometers downwind, superimposing with wakes from other wind farms. At 17:17, the SAR 10-m wind speed at the FINO 1 site is about 8 m s$^{-1}$ , and the WRF outputs, both from the Fitch and the EWP schemes, are also about 8 m s$^{-1}$. Note that in a short distance downwind of the wind farms, the surface wind at 10 m from the Fitch scheme suggests a slight speedup, see the brighter color in the farm wake shadows in Fig. 1b and the white color in Fig. 9b, c and d. This speedup is a phenomenon that deserves further investigation (Djath et al., 2018), but it is beyond the scope of this study. Moreover, all four wind farm simulations in Fig. 9 suggest a presence of reduced wind speed in front of the farms. This is referred to as global blockage effect (Bleeg et al., 2018; Schneemann et al., 2021). Such an effect is most obvious for wind farms with free upstream flow and it is more obvious in the Fitch simulations.

The transect-flight data over transect-a are plotted in Fig. 10, for wind speed at 250 m (a,c) and TKE (b,d) at 250 m, respectively. To improve the visibility of individual model scenarios, Fig. 10a and b compare measurements, NWF, EWP and Fitch-off, and Fig. 10c and d compare measurements, Fitch-on-1, Fitch-on-025 and Fitch-off. There are altogether six flights over transect-a between approximately 14:20 and 16:10 (Table 1), each lasted approximately 10 minutes. We averaged the flight data over a spatial distance of both 2 km (same as the WRF spatial resolution) and 1.5 km (same as in S2020). The results of the two averaging distances are similar; the one using 1.5 km provides slightly more fluctuation. In Fig. 10 we only show the results using 2 km. The corresponding model data at 250 m over transect-red in Fig. 3 from 14:00 to 16:00, covering the flight periods, are plotted. The modeled data are 10-min instantaneous values, plotted every half an hour. Based on Fig. 10a and c, compared to the ambient flow, there is a deficit in wind speed at 250 m above the wind farm Godewind 1, due to the wind farm effects. Such a wind deficit is almost 3 m s$^{-1}$ in the flight data, and it is only about 2 m s$^{-1}$ in the modeled data using EWP, about 3 m s$^{-1}$ in Fitch-on-1, slightly smaller reduction for Fitch-off and even smaller reduction for Fitch-adv-0.25.

As a result of the wind farm parameterizations, above the wind farm high TKE values are observed when compared with ambient values, which are almost zero (Fig. 10b and d). When no wind farms are included in the modeling, there is no systematic difference in TKE across transect-a, see the black curves in Fig. 10b. Parameterizations of the wind farms result in increased TKE over the farm. At 250 m, the Fitch-on-1 scheme provides TKE inside the range of the measured values, though with large underestimation at the southern edge of the farm and with comparable magnitude at the northern edge of the farm. The profiles at point A (Fig. 7b), which is approximately in the center of transect-a and transect-red (Fig. 3), show that the TKE values are highest close to hub height and decrease above it. At point A, the TKE value simulated by the Fitch-on-1 and Fitch-off schemes are about $2.1 - 2.3$ m$^2$ s$^{-2}$ at rotor top height, which is about 40% higher than the values at 250 m. For transect-a, the TKE values from the EWP scheme are considerably smaller, being only about $1/3 - 1/4$ of the values from the Fitch scheme, and are thus significantly underestimated compared to the flight measurements. The overall TKE magnitudes across the transect from Fitch-on-1 and Fitch-off are also comparable. Due to the activated TKE advection, Fitch-on-1 provides a more even-distribution across the transect compared to Fitch-off. Fitch-on-025, which also shows the even distribution of TKE across the transect, has significantly lower values of TKE compared to Fitch-on-1, Fitch-off and measurements.

One can notice the speedup in the flow in the flight measurements on the southern edge of the farm in Fig. 10a and c, as also pointed out in S2020; see the bump of wind speed at 250 m at latitude before 54°N. WRF does not capture this phenomenon with either scheme. The abrupt increase in TKE in the same area (Fig. 10b) is likely related to this flow acceleration and is also missing in the WRF results.

The vertical distributions of the wind speed and TKE along transect-a are shown in contour lines in Fig. 11 as latitude (from south to north) versus height, for 15:30 for both EWP and three Fitch simulations, as an example. The LLJs are visible here in Fig. 11 (left column) with the wind speed maximum height between ∼200 and 500 m. The largest difference in the wind speed between the Fitch and EWP schemes is over the farm beneath the rotor top, with Fitch simulating on average larger wind speed reductions from the wind farm wake effect (Fig. 11a and c,e,g).

The right column in Fig. 11 shows that, the largest TKE values are located at a height between hub height and rotor top. The three Fitch simulations correspond to several times larger TKE values than the EWP scheme, with the largest difference

over the wind farm area. An increase in TKE is notable up to double the height of the rotor top. The maximum TKE values are largest with Fitch-off, and they are slightly smaller with Fitch-on-1 and significantly smaller with Fitch-on-0.25.

Outside the farm area and even in the wake region, the EWP and Fitch-off schemes provide similar wind speed and TKE values, see for example Fig. 12, which is for the transect-purple (Fig. 3b, with longitude approximately 7.2°E). This resemblance is not given when turbine-induced TKE is advected downwind of the farm such as in Fitch-on-1 and Fitch-on-0.25. Compared to Fitch-off, in the far wake region, TKE from Fitch-on-1 and Fitch-on-0.25 are larger above 200 m, which is above the Godewind 1 turbine hub height, see the right column of Fig. 12.

These characteristics can also be seen in the horizontal spatial distribution of the wind speed (Fig. 13) and TKE (Fig. 14) at 250 m over the wind farm cluster with the Godewind 1 farm in the domain center. The patterns of spatial distribution of wind speed are consistent with the four plots in Fig. 13, though EWP shows overall smaller wind speed reduction downwind of the farm and Fitch-on-1 corresponds to overall larger reduction, when being compared to the rest. For the TKE, when TKE-advection is turned off, farm-induced TKE is mostly above the wind farm (Fig. 14b, Fitch-off). When the advection is turned on, the high-TKE values are transported downwind of the farm in the mean wind direction, in Fig. 14c (Fitch-on-1) and d (Fitch-on-0.25), respectively. TKE in the EWP scheme, as a function of the wind shear, follows the mean flow. When the TKE-advection is turned off, EWP-off provides similar spatial distribution to Fig. 14a, only with the maximum TKE slightly less-away from the Godewind 1 farm (not shown).

## 4 Discussions and conclusions

For the first time, the calculations of both wind speed and TKE from two explicit wind farm parametrization schemes (Fitch and EWP) in WRF are compared and verified through a case study, thanks to the open access high frequency flight data over and around the wind farm Godewind 1 (Bärfuss et al., 2019a) and FINO 1 measurements. This study thus complements S2020 where only the Fitch scheme was used to model the wind farm wake. Additionally, we used a WRF version, where the tubine-induced TKE is correctly integrated in and advected with the overall TKE field, following Archer et al. (2020).

The farm wake effect is discussed here with a variety of measurements: in the FINO 1 mast measurements shown as vertical wind-profile and time series, in the SAR 10-m wind speed shown as spatial distribution and in the flight data shown as crosswind farm transect distribution of wind speed and TKE at 250 m. The WRF modeling with the two farm parametrization schemes EWP and Fitch captures these observed farm wake effect consistently in terms of wind speed, but with some noticeable differences. In the vertical wind-profiles at FINO 1, when compared to the EWP scheme, the wind speed deficit due to the wind farm wakes using the Fitch scheme is more centered and more pronounced around the hub height and rotor area, visible as a kink in the profile. In the EWP scheme, due to the subgrid-scale vertical wake expansion, the wind deficit is more spread and smooth over the rotor area. The larger wind speed deficit in association with the Fitch scheme beneath the rotor top height is also visible over the wind farm and in the wake areas. The flight data suggests a flow acceleration on the southern side of the wind farm Godewind 1 accompanying the flow from the southwest. This acceleration is however not captured in WRF. It is expected that a high fidelity model is needed to investigate further this feature.

Even though the modeled wind speeds are comparable using the two schemes, the results on TKE are significantly different. The Fitch scheme, having TKE contributions from both the shear and an explicit term related to the turbine power and thrust coefficients, provides TKE values several times larger than those from the EWP scheme. In addition, WRF misses the flow acceleration south to the farm and it underestimates TKE in the adjacent wind farm area. At the northern part of the farm, the modeled TKE values using the Fitch scheme are of comparable magnitudes with respect to to the measurements. TKE values from the EWP scheme are significantly underestimated. This suggests that turbine-induced TKE does not only develop from the shear, as assumed in the EWP scheme, but instead an explicit source is required.

Most recent studies in the literature recommended deactivating TKE-advection when using the Fitch scheme, based on simulations with WRF containing a code bug that incorrectly treated the turbine-induced TKE in the advection scheme and incorrect neglect of electro-mechanical losses as pointed out in (Archer et al., 2020). Archer et al. (2020)'s analysis suggests that the two issues interact with each other, causing compensating errors. This generated TKE values in realistic range, which might be the reason that these issues haven't been identified before. S2020 found that, with the bug, their simulation is in better agreement with measurements with advection turned off.

All results presented here are from the bug-fixed version of WRF, following Archer et al. (2020). Accordingly, we tested two correction factors, $\alpha = 0.25$ and 1. Archer et al. (2020)'s study recommended using $\alpha$ less than 1, e.g. 0.25, according to their verification with Large Eddy Simulations. This is is however not supported by the current study. Using $\alpha = 0.25$ does not always improve the results, as can be seen in the comparison with measurements from FINO 1, or with the flight data over the Godewind 1 farm. With TKE-advection turned on in Fitch, TKE is no longer concentrated above the wind farm but advected with the mean flow, similar to EWP in which the shear-induced turbulence naturally follows the mean flow advection. With TKE-advection turned on and $\alpha = 1$, the results of mean wind speed field are quite similar to that with TKE-advection turned off. While using $\alpha = 0.25$ gives similar spatial distribution of wind speed and TKE, it gives quite different magnitudes of speed-reduction as well as TKE values. Due to lack of measurements, it remains inconclusive how much the inclusion of farm-induced-TKE advection improves the results and what are the correct $\alpha$ values to use. More measurements and more meteorological conditions are needed for further investigation.

Since most studies in the literature using the Fitch scheme are affected by the code bug, here we also briefly examine some results from WRF before the bug-fix. Fig. 15 shows the profiles at point A, similar as Fig. 7, but now also includes a simulation with WRF-Fitch with the bug and advection turned on (the only option for $\alpha$ is 1 in that version), the results are very different both for wind speed and TKE, compared to the bug-fixed version with $\alpha = 1$. This result suggests that the effect of this bug is not negligible.

The impact of deactivated TKE-advection was also tested in connection with the simulation with no farms (NWF-off) and the EWP scheme (EWP-off). Compared to NWF and EWP, the differences in the wind field are not systematic but shown to be random noises with NWF-off and EWP-off (not shown). One can notice that TKE related to turbine-affected shear travels slightly further in EWP than EWP-off. The overall effect on the wind speed and TKE is marginal and do not change the findings in the study.

The simulations here also suggest the presence of a global blockage effect when the flow approaches the wind farms. Though future studies are required to understand if such an effect is accurately captured, and which scheme describes such an effect better.

The studied case becomes even more interesting due to the presence of LLJs, as LLJs are a common phenomenon in this area of the North Sea. Numerical modeling studies of wind energy usually address the rich resource in connection with LLJs, though few have included a wind farm wake effect. This case study shows an overestimation of the 100-m wind speed by approximately 1.45 m s$^{-1}$ at FINO 1 when ignoring the wind farm effect in the WRF modeling, accounting $11-20\%$ of the mean wind speed during this simulation period. At the same time, measurements and modeling at FINO 1 suggest that the wind speed profile in the presence of LLJs is modified by the presence of wind farm wakes. The wind farm wakes lead to a reduced wind speed up to the jet nose, a higher jet nose and a higher wind shear beneath the jet nose. This upward shift of the jet nose in the presence of a wind farm was also modelled in different LES studies (Sharma et al., 2017; Abkar et al., 2016).

## 5   Conclusions

It is important to take the wind farm wake effect into account when calculating LLJ wind speeds in areas with wind farms. LLJ structures are affected by the presence of wind farms. The WRF model with both the Fitch and the EWP schemes can capture the wind speed field rather well and consistently. It remains inconclusive which scheme is better at describing the wind field, as sometimes the EWP scheme outperforms the Fitch scheme, and some other times, it is the other way around. It also remains inconclusive which correction factor should be used in connection with the turbine-induced TKE generation in the Fitch scheme: we only tested two factors (1 and 0.25) here and we observe a better performance when using $\alpha = 1$ than $\alpha = 0.25$, which does not support the conclusion from Archer et al. (2020). TKE from the EWP scheme is significantly underestimated compared to the flight measurements. This suggests that an explicit turbine-induced source of TKE should be included in addition to the shear-generated TKE. Neither scheme can capture the flow acceleration along the farm edge.

This case study shows typical features of the wind farm wakes in the presence of LLJs, using the most-used modeling approaches. It raises issues that have not been addressed in the literature, namely the interaction of wind farm wakes and LLJs. It also clearly shows the need for improvements of turbine-induced TKE calculations using the wind farm parameterizations in WRF. This study therefore serves as a start for a more systematic study of similar conditions.

*Code and data availability.*   The WRF model code is publicly available at https://github.com/wrf-model/WRF. The WRF input files and the source code for the wind farm parameterizations of Volker et al. (2015) as well as of Fitch et al. (2012) with bugfix by Archer et al. (2020) are permanently indexed at https://doi.org/10.5281/zenodo.4668613. Updates for EWP will be made available on https://gitlab.windenergy. dtu.dk/WRF/EWP. The SAR data are available from https://satwinds.windenergy.dtu.dk/. The FINO 1 measurements can be assessed from http://fino.bsh.de/. The flight data are available on https://doi.org/10.1594/PANGAEA.902845. The OSTIA data is available from http:// my.cmems-du.eu/motu-web/Motu. ERA5 data was downloaded from https://doi.org/10.24381/cds.bd0915c6. Data and scripts required to reproduce the analysis in this study are also shared in the above Zenodo-record at https://doi.org/10.5281/zenodo.4668613.

*Author contributions.* XL outlined the manuscript. XL and JF ran the simulations, performed the data analysis and wrote the draft.

395 *Competing interests.* The authors declare that they have no conflict of interest.

*Acknowledgements.* This study is supported by the ForskEL/EUDP OffshoreWake project (PSO-12521/EUDP 64017-0017). We thank the open source platform https://doi.pangaea.de/10.1594/PANGAEA.902845 for the flight data. We thank the Federal Maritime and Hydrographic Agency (BSH) for providing measurements at the FINO 1 station. We thank our colleagues Jake Badger, Andrea Hahmann and Rogier Floors for discussions. Data processing and visualization for this study was in part conducted using the python programming
language and involved use of the following software packages: NumPy (van der Walt et al., 2011), pandas (McKinney, 2010), xarray (Hoyer and Hamman, 2017), Matplotlib (Hunter, 2007). The colours for the line plot have bee selected through the 'Color Cycle Picker' https://github.com/mpetroff/color-cycle-picker. The authors are grateful for the tools provided by the open-source community.

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

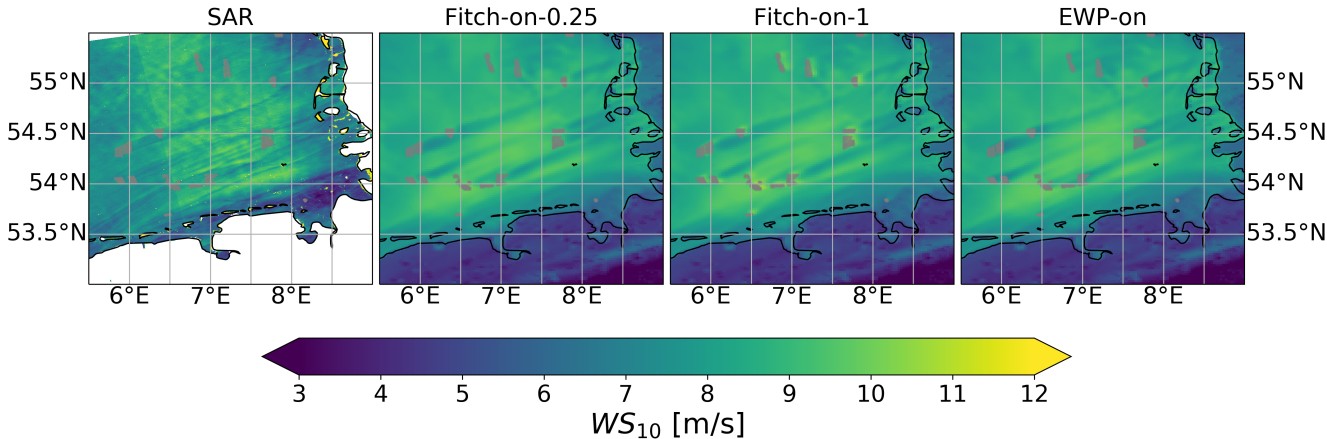

**Figure 1.** Wind speed (ms$^{-1}$) at about 10 m height (a) from SAR at 17:17 UTC on 14 Oct 2017 and (b-d) from WRF at 17:10 on 14 Oct for different scenarios as in Table 3. The satellite data in (a) are taken from https://satwinds.windenergy.dtu.dk/.

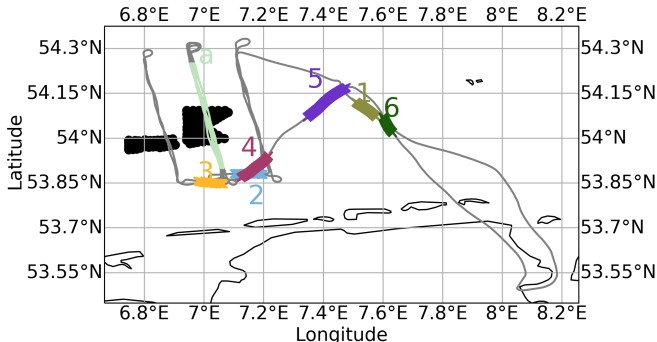

**Figure 2.** Flight tracks on 14 Oct 2017. Track labeled with "a" provides transect-flight data over the wind farm at about 250 m. Tracks labeled with numbers 1 to 6 provide the profile-flight data (see also Table 1). The flight track has been extracted from Bärfuss et al. (2019b).

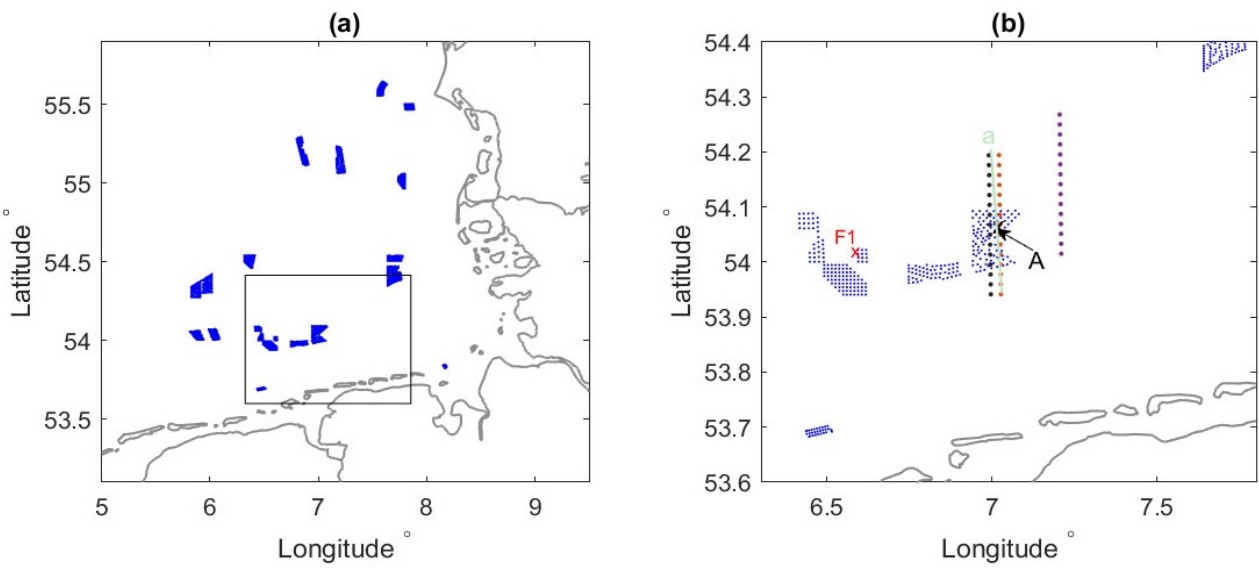

**Figure 3.** (a). Wind farm clusters that are included in the WRF modeling. The box includes the wind farms shown in (b) which is a close-up of (a), where the two consecutive rows over the Godewind 1 farm are WRF grid points (black and red), the flight legs are in between the two row of WRF grid points (transect labeled "a" in green as in Fig. 2 ) and one more row down wind (purple). Also marked are the location of the FINO 1 mast (F1) and point A on transect-a.

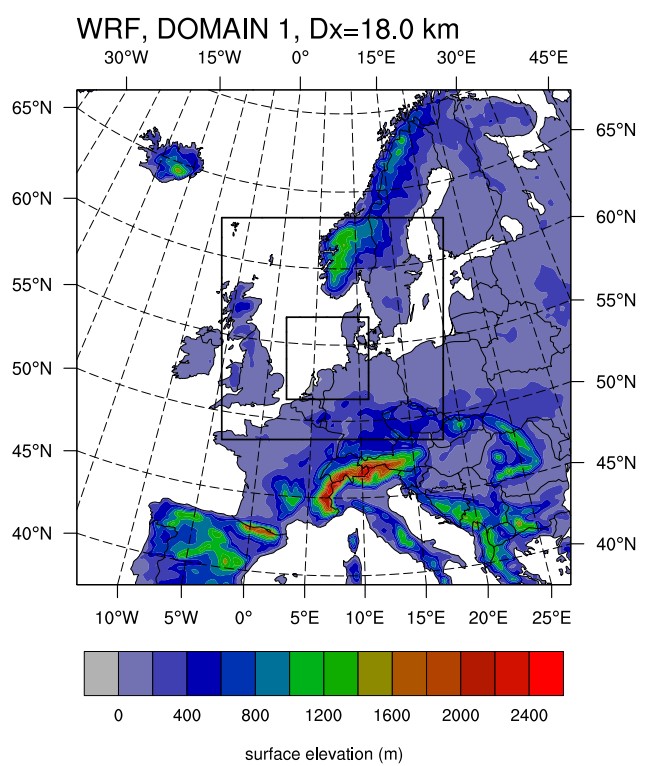

**Figure 4.** The three nested model domains in the WRF modeling, colors show topography.

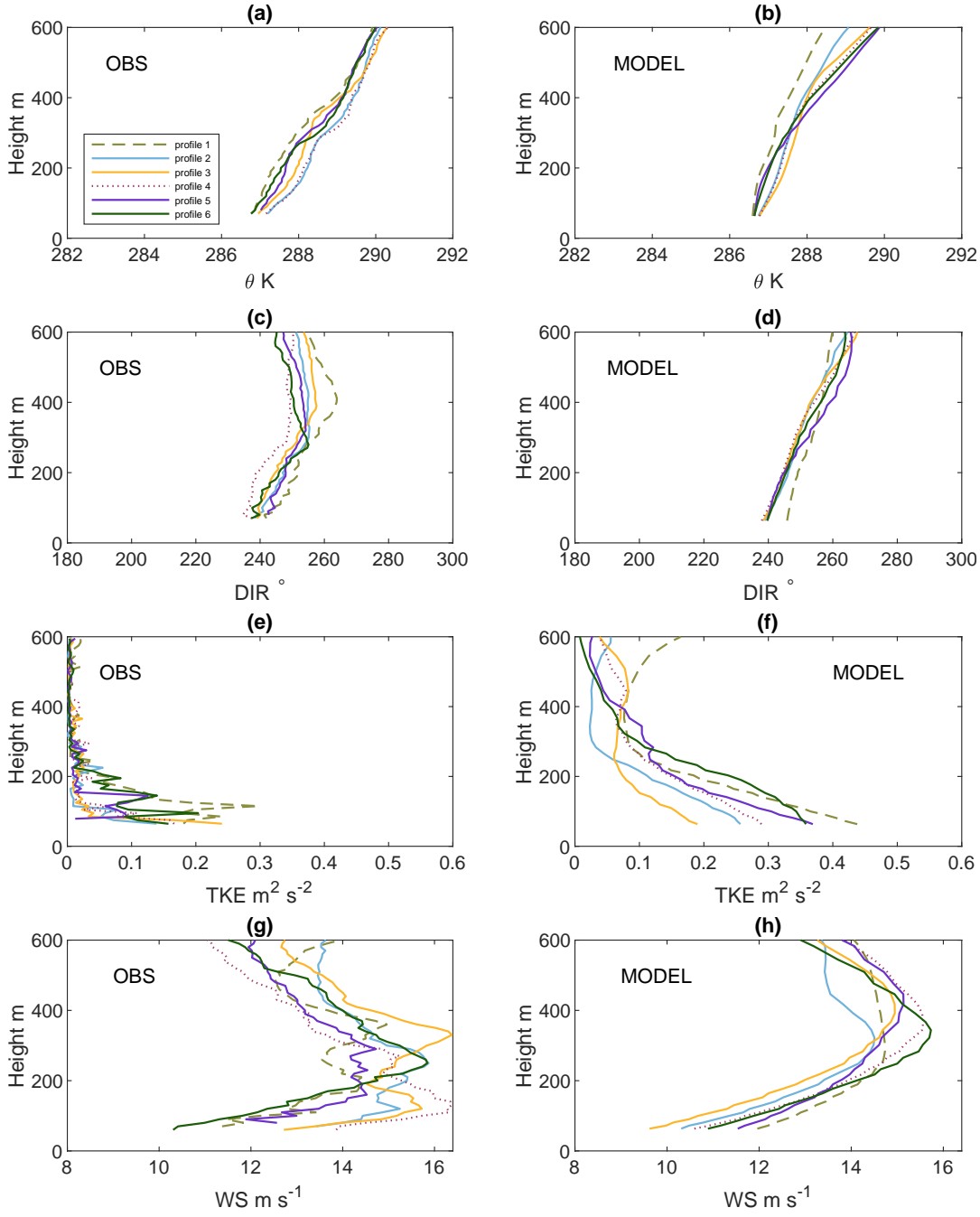

**Figure 5.** Vertical profiles of potential temperature (a, b), wind direction (c, d), TKE (e, f) and wind speed (g, h), observed (OBS) and modeled (MODEL) at the center positions of profile-flight 1 to 6. Note that the modeled data are from EWP scheme.

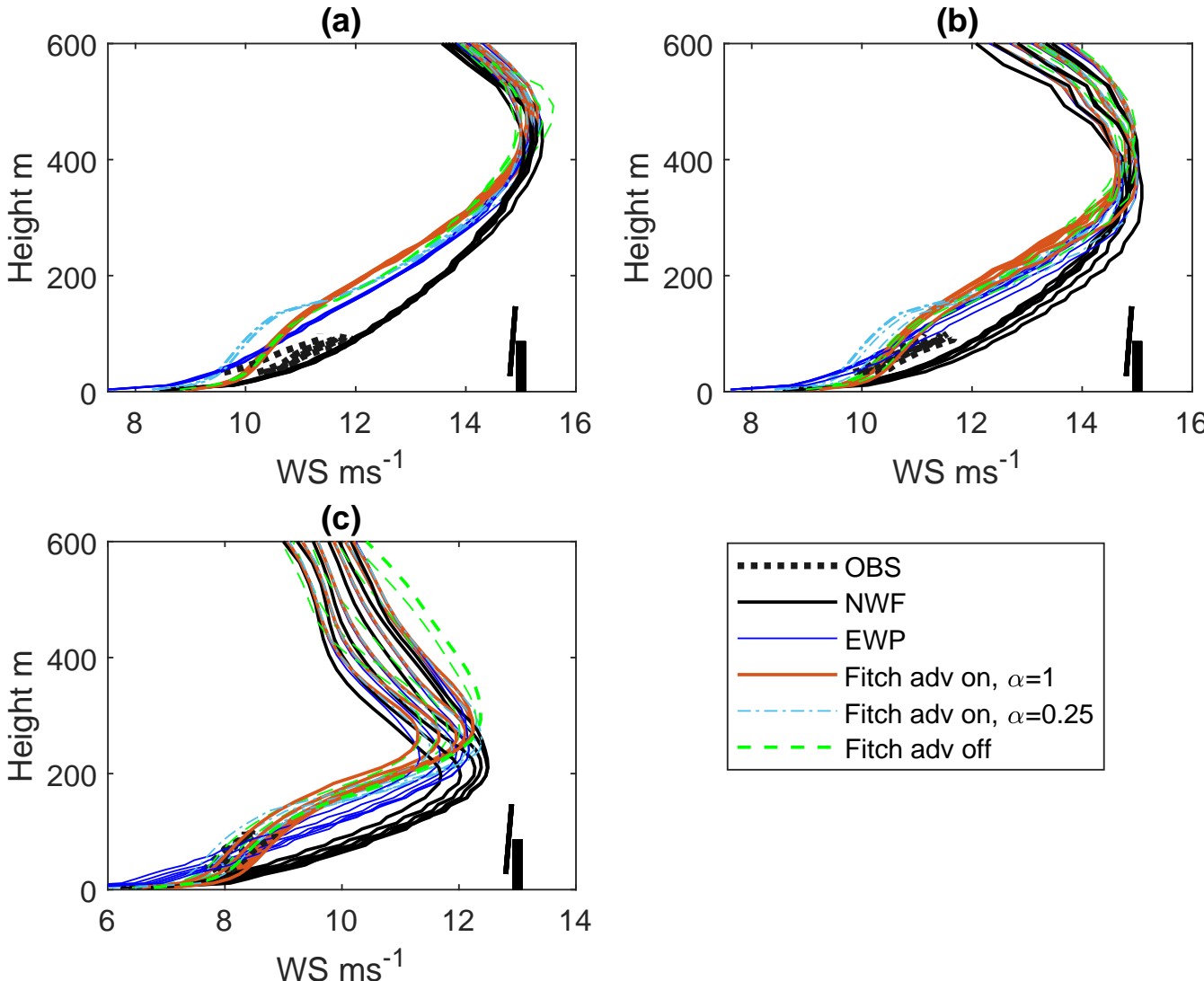

**Figure 6.** Wind profiles measured (OBS) and modeled (NWF, EWP, Fitch-on-1, Fitch-on-0.25, Fitch-off) at FINO 1 station on 14 Oct. (a) 10-min profiles between 14:00 and 15:00; (b) 10-min profiles from 15:00-16:00; (c) 10-min profiles from 20:00-21:00. The corresponding turbine hub height and the rotor area from the upwind farm Borkum Riffgrund 1 are illustrated in black

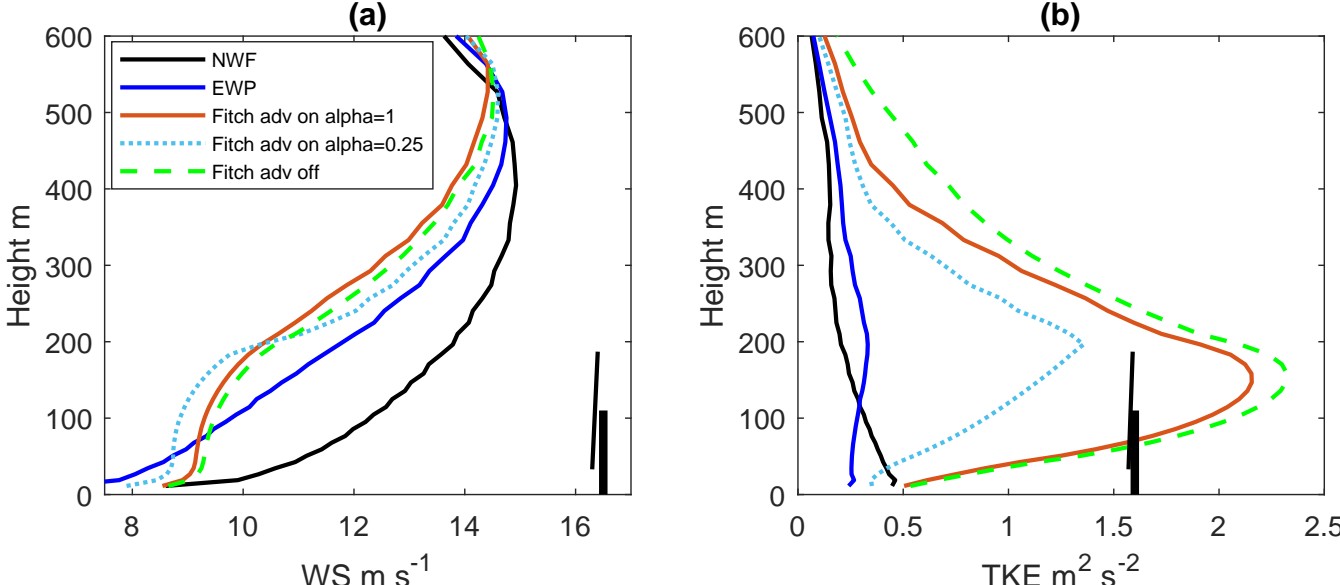

**Figure 7.** Modeled vertical profiles at Point A (see Fig. 3) over the Godewind 1 wind farm at 15:00 on 14 Oct, together with the transect-flight-4 data at 250 m (15:01 - 15:11) (a) wind speed; (b) TKE. The corresponding turbine hub height and the rotor area are illustrated in black.

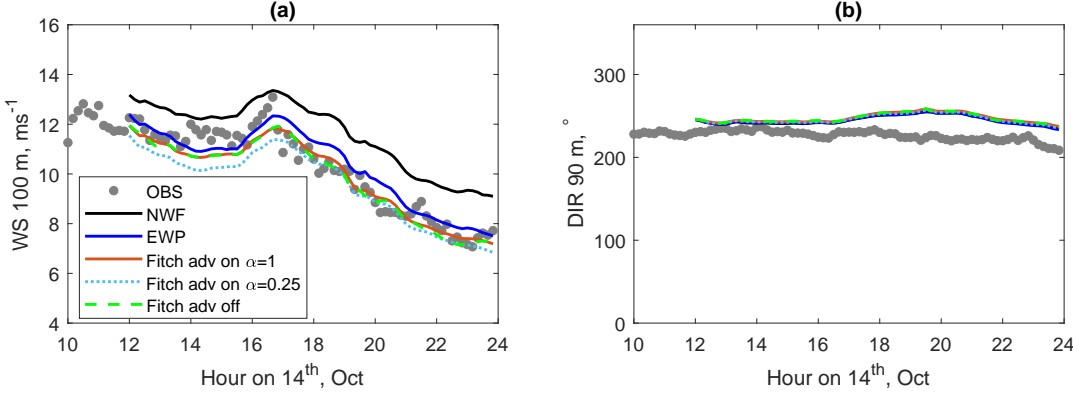

**Figure 8.** Measured and modeled time series at FINO 1 on the 14 Oct 2017: (a) wind speed at 100 m; (b) wind direction at 90 m, using Fitch and EWP schemes, as well as no farms option (NWF).

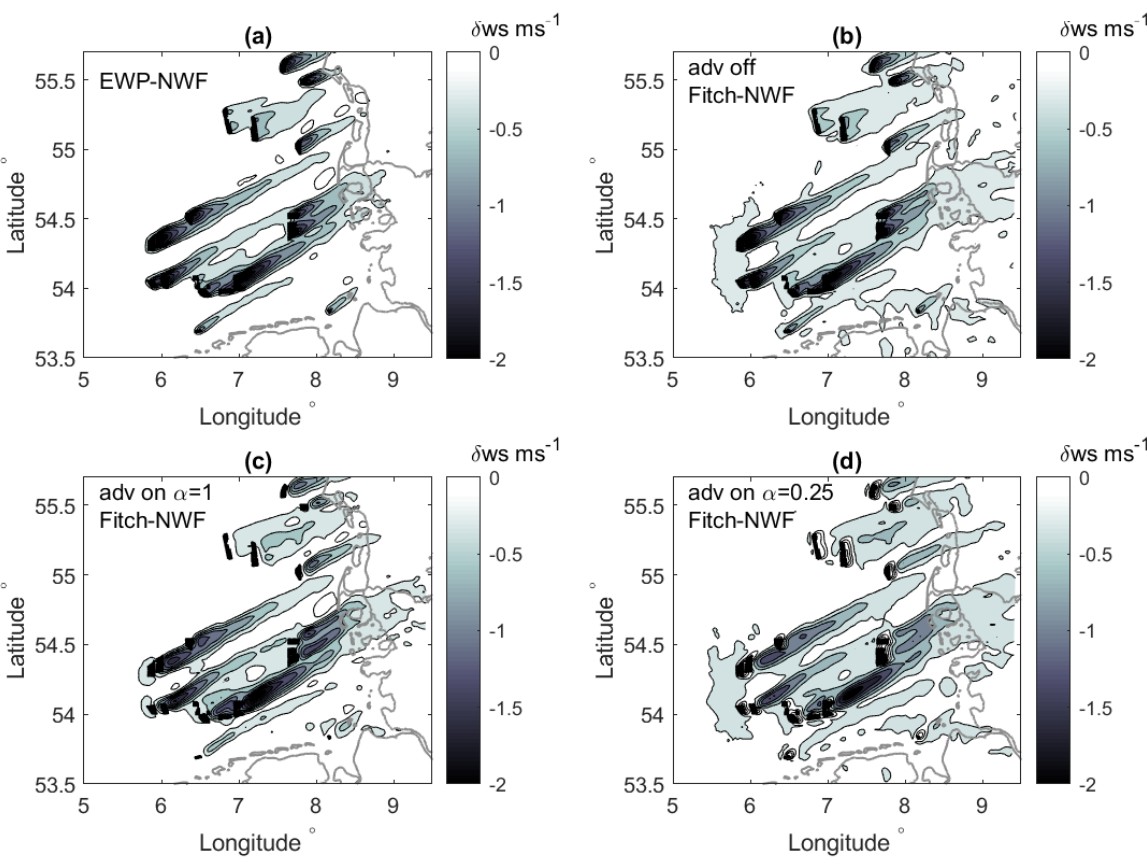

**Figure 9.** Wind speed deficit (m s$^{-1}$) at about 10 m 17:00 on 14 Oct 2017, corresponding to the SAR image as in Fig. 1. (a) deficit between EWP and NWF; (b) deficit between Fitch-off and NWF; (c) deficit between Fitch-on-1 and NWF; (d) deficit between Fitch-on-025 and NWF.

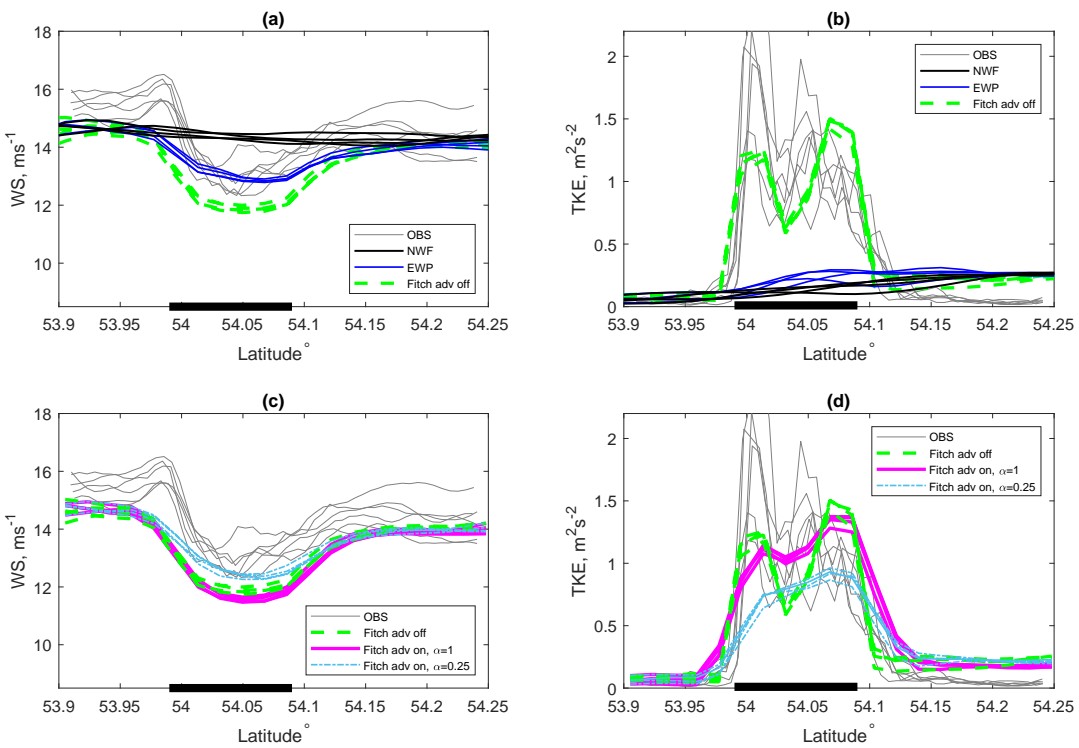

**Figure 10.** (a, c) Transect-a distribution of wind speed at 250 m as a function of latitude between 14:00 to 16:00 on 14 Oct 2017, observed (OBS) and modeled (every 30-min). (b, d) Similar to (a, c) but for TKE. The observed values are the flight data averaged over a distance of 2 km. The modeled values are from the use of the Fitch, the EWP and no wind farm (NWF) schemes. The wind farm is indicated on the $x-$axis with a thick black line.

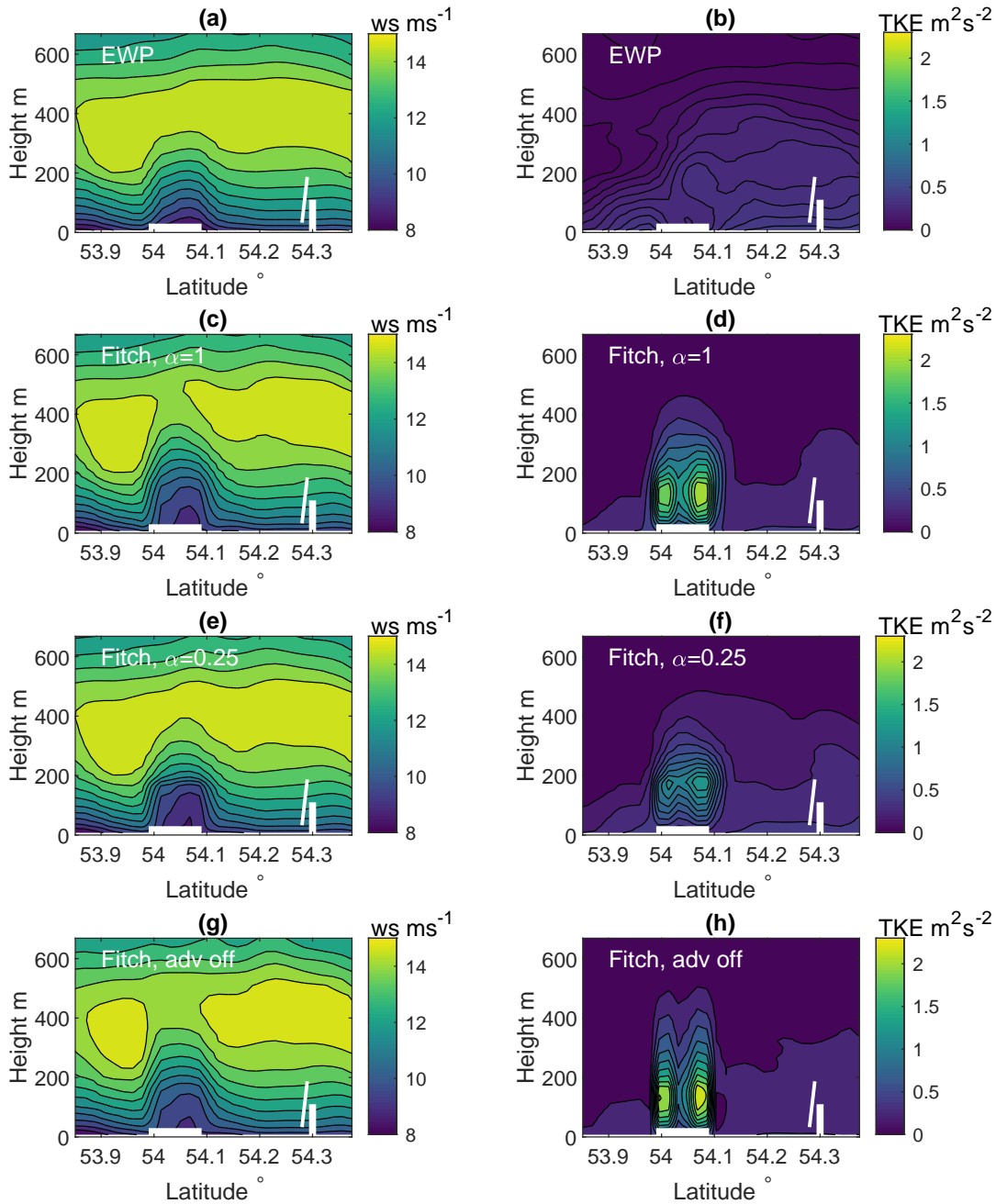

**Figure 11.** Distribution of wind speed (left column) and TKE (right column) over the transect-red (at longitude 7.02°E) at 15:30 on 14 Oct 2017, $x$-axis: south-north and $z$-axis: height. Row-1: EWP. Row-2: Fitch-on-1. Row-3: Fitch-on-025. Row-4: Fitch-off. The wind farm is indicated on the $x-$axis with a thick white line and the corresponding turbine is illustrated in white.

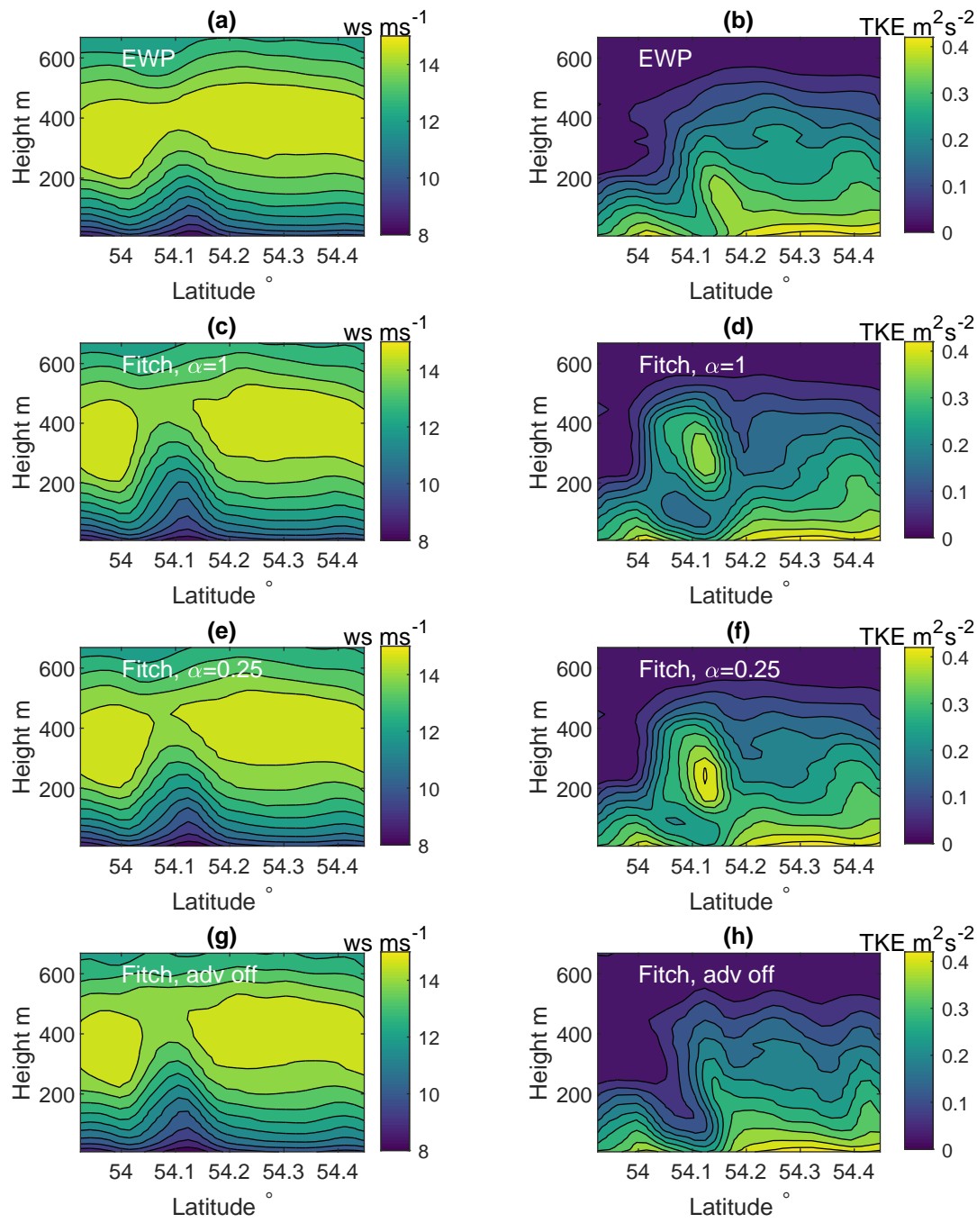

**Figure 12.** Similar to Fig. 11, but for transect-purple (at longitude 7.2°E) downwind of wind farms at 15:30 on 14 Oct.

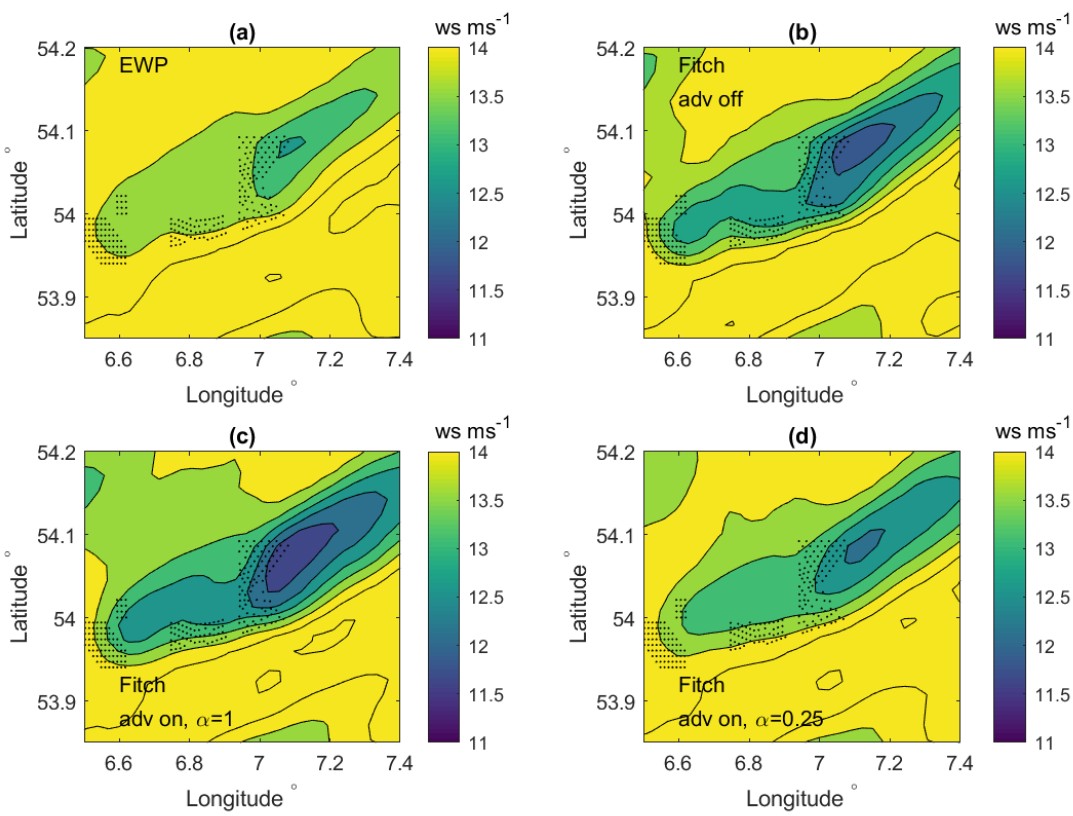

**Figure 13.** Spatial distribution of wind speed at 250 m from WRF model at 20171014 15:30 (a) using EWP scheme; (b) using Fitch scheme, advection of (Fitch-off); (c) using Fitch scheme, with advection on and $\alpha = 1$ (Fitch-on-1); (d) using Fitch scheme, with advection on and $\alpha = 0.25$ (Fich-on-0.25)

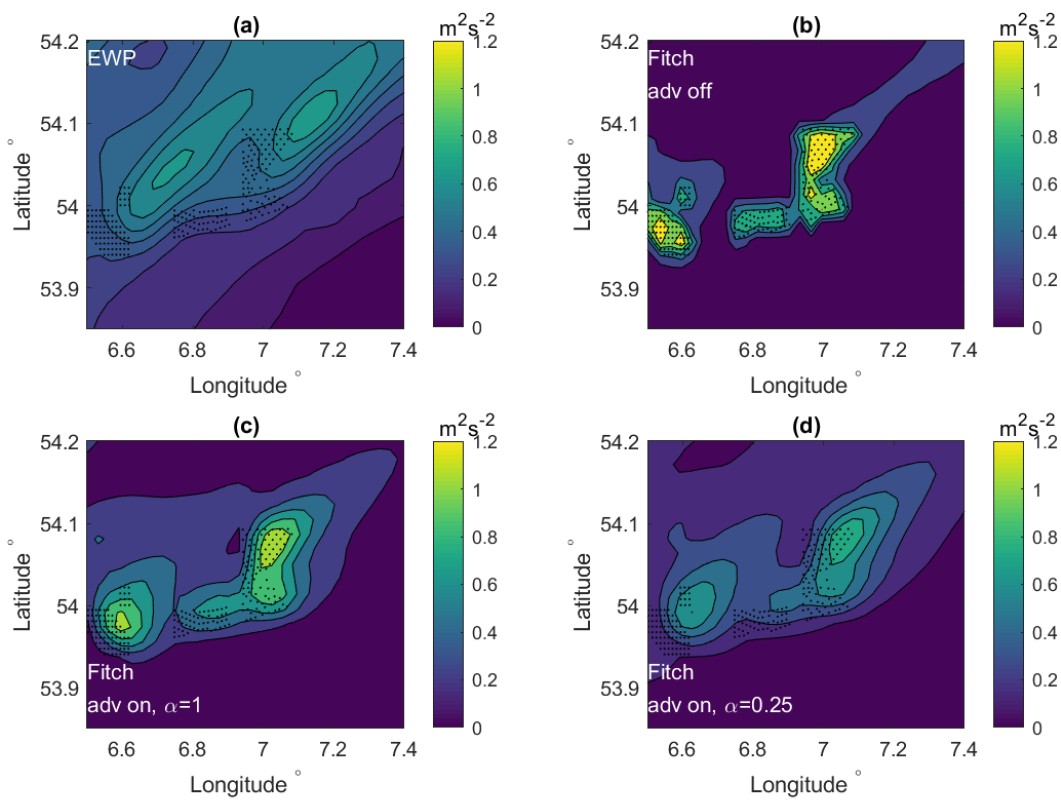

**Figure 14.** Spatial distribution of TKE at 250 m from WRF model at 20171014 15:30 (a) using EWP scheme; (b) using Fitch scheme, advection of (Fitch-off); (c) using Fitch scheme, with advection on and $\alpha = 1$ (Fitch-on-1); (d) using Fitch scheme, with advection on and $\alpha = 0.25$ (Fich-on-0.25)

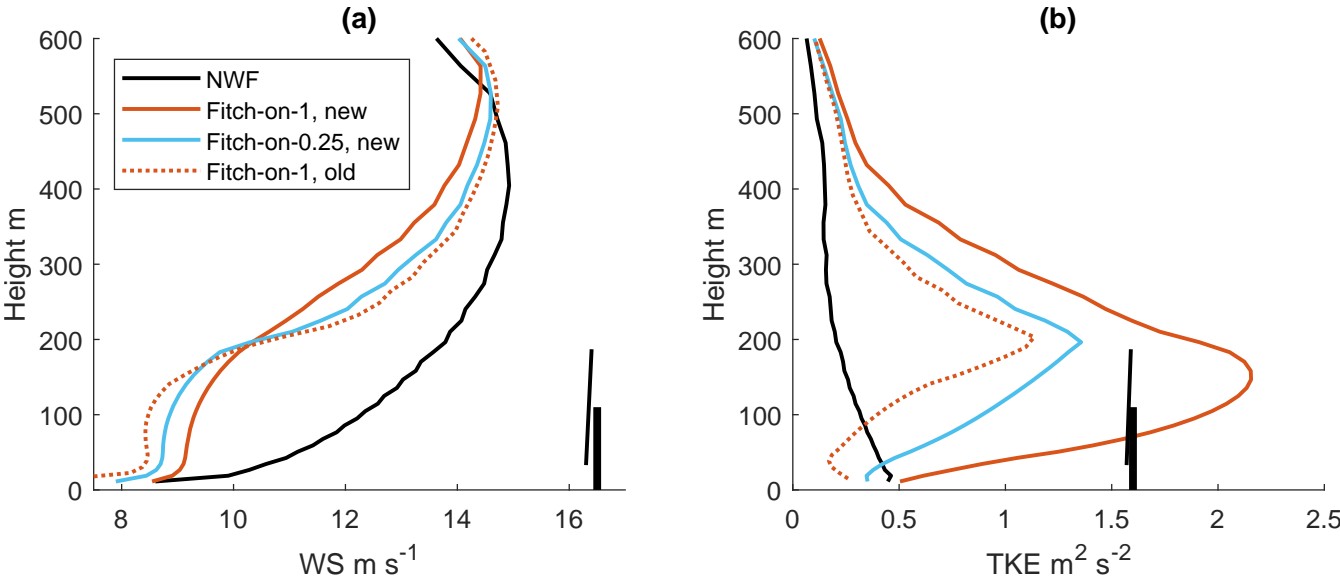

**Figure 15.** Similar to Fig. 7, except for EWP, Fitch-on-1, Fitch-on-0.25 and Fitch-old.

**Table 1.** Time of the profile-flights (see track 1-6 in Fig. 2) and transect-flights (see Fig. 2 transect-a).

| Flight nr. | start | end |
|---|---|---|
| profile 1 | 2017-10-14 13:22:59.700 | 2017-10-14 13:25:08.410 |
| profile 2 | 2017-10-14 14:14:41.600 | 2017-10-14 14:17:10.470 |
| profile 3 | 2017-10-14 15:13:31.260 | 2017-10-14 15:15:53.240 |
| profile 4 | 2017-10-14 16:10:21.570 | 2017-10-14 16:12:49.110 |
| profile 5 | 2017-10-14 16:16:35.240 | 2017-10-14 16:19:55.230 |
| profile 6 | 2017-10-14 16:23:22.060 | 2017-10-14 16:25:05.150 |
| transect 1 | 2017-10-14 14:20:50.860 | 2017-10-14 14:30:12.370 |
| transect 2 | 2017-10-14 14:34:41.180 | 2017-10-14 14:44:37.520 |
| transect 3 | 2017-10-14 14:48:27.970 | 2017-10-14 14:57:43.640 |
| transect 4 | 2017-10-14 15:01:38.120 | 2017-10-14 15:11:34.970 |
| transect 5 | 2017-10-14 15:45:01.130 | 2017-10-14 15:54:05.160 |
| transect 6 | 2017-10-14 15:58:29.630 | 2017-10-14 16:08:34.810 |

**Table 2.** WRF parameterisation, boundary conditions and forcing data employed for the performed simulations.

| Category | Subcategory | Details (option number) |
|---|---|---|
| Schemes | PBL | MYNN (Nakanishi and Niino, 2009) |
| | Surface layer | Monin-Obukhov similarity |
| | Microphysics | New Thompson et al. scheme (Thompson et al., 2004) |
| | Radiation | RRTMG scheme (Iacono et al., 2008) |
| | Cumulus parameterisation | Kain-Fritsch scheme on domain 1 (Kain and Fritsch, 1993) |
| Boundary and forcing data | Dynamical forcing | ERA-5 on pressure levels every 6 hours |
| | Land use data | CORINE from 2017 |
| | Sea surface temperature | OSTIA (Donlon et al., 2012) |
| | Land surface model | NOAH-LSM |

**Table 3.** Overview on performed simulations.

| Denotation | Wind Farm Parameterisation | TKE advection |
|---|---|---|
| EWP | EWP | on |
| EWP-off | EWP | off |
| Fitch-on-0.25 | Fitch | on; $\alpha = 0.25$ |
| Fitch-on-1 | Fitch | on; $\alpha = 1$ |
| Fitch-off | Fitch | off |
| Fitch-on-old | Fitch | on (before bugfix) |
| NWF | No | on |
| NWF-off | No | off |

**Table 4.** Wind farm details for all simulated wind farms. Note that for some wind farms the turbine models do not correspond to the actually installed one, since thrust and power curves where not available for all turbines: here turbines marked with [1] are scaled from NREL 5 MW turbine and turbines marked with [2] are scaled from DTU 10 MW turbine.

| Wind farm | Turbines | Turbine Model | Hub Height [m] | Rotor top [m] | Wind Farm Area [km$^2$] |
|---|---|---|---|---|---|
| Alpha Ventus | 12 | M5000-116[1], Senvion_5M[1] | 90.0, 92.0 | 148.0, 155.0 | 4 |
| Amrumbank West | 80 | SWT-3.6-120 | 88 | 148 | 30 |
| BARD Offshore | 80 | M5000-116[1] | 90 | 148 | 59 |
| Borkum Riffgrund 1 | 78 | SWT-4.0-120 | 89.5 | 149.5 | 36 |
| Butendiek | 80 | SWT-3.6-120 | 88 | 148 | 31 |
| Gemini | 150 | SWT-4.0-130 | 95 | 160 | 68 |
| Global Tech I | 80 | M5000-116[1] | 90 | 148 | 40 |
| Gode Wind 1 | 55 | SWT-6.0-154_110 | 110 | 187 | 40 |
| Gode Wind 2 | 42 | SWT-6.0-154_110 | 110 | 187 | 29 |
| Horns Rev I | 80 | V80-2.0 | 67 | 107 | 21 |
| Horns Rev II | 91 | SWT-2.3-93 | 68.3 | 114.8 | 33 |
| Meerwind Süd/Ost | 80 | SWT-3.6-120 | 88 | 148 | 40 |
| Nordsee One | 54 | 6.2M126_90[2] | 90 | 153 | 30 |
| OWP Nordergründe | 18 | 6.2M126_84[2] | 84 | 147 | 3 |
| OWP Nordsee Ost | 48 | 6.2M126_95[2] | 95 | 158 | 36 |
| OWP Veja Mate | 67 | SWT-6.0-154 | 106 | 183 | 51 |
| Offshore Windfarm DanTysk | 80 | SWT-3.6-120 | 88 | 148 | 65 |
| Offshore Windfarm Sandbank | 72 | SWT-4.0-130 | 95 | 160 | 47 |
| Offshore Windpark Riffgat | 30 | SWT-3.6-120 | 88 | 148 | 6 |
| Trianel Windpark Borkum | 40 | M5000-116[1] | 90 | 148 | 23 |

**Table 5.** Differences between measurements and simulated wind speed time series at 100 m at FINO 1 between 12:00 and 24:00 on 14 Oct, in terms of mean deficit, standard deviation of the difference and absolute difference. Positive values mean that the measured values are larger.

| Denotation | $\langle \Delta U \rangle$ ms$^{-1}$ | $STD$ ms$^{-1}$ | $\langle |\Delta U| \rangle$ ms$^{-1}$ |
|---|---|---|---|
| EWP | -0.21 | 0.57 | 0.50 |
| Fitch-on-0.25 | 0.59 | 0.56 | 0.65 |
| Fitch-on-1 | 0.22 | 0.50 | 0.43 |
| Fitch-off | 0.30 | 0.46 | 0.44 |
| NWF | -1.45 | 0.59 | 1.45 |