# Peer review of "A case study of wind farm effects using two wake parameterizations in WRF (V3.7.1) in the presence of low level jets"

_Geoscientific Model Development, 2020_

## Referee Comment (RC1) · Simon Siedersleben (Referee) · 2 Feb 2021

Reviews corresponding to the article: "A case study of wind farm effects using two wake parameterizations in WRF (V3.7) in the presence of low level jets"

Authors: Xiaoli G. Larsen and Jana Fischereit

**General comment:**

In general, this work makes an important contribution to the wind energy resource assessment. The two most prominent wind farm parametrizations are evaluated by the use of airborne measurements and are compared against each other. This is important as the wind farm parameterization of Volker et al. (2015) has been used in a recently published report, indicating that wakes in the German Bight could have a massive impact on the energy harvesting in the future, although this parameterization was so far only poorly evaluated for offshore wind farms. However, three major points have to revised in this manuscript:

First of all, the description of the TKE coefficient ( $C_{TKE}$ ) that determines the amount of TKE added by the wind farm parameterization is highly confusing. In this manuscript they authors describe  $C_{TKE}$  as a constant value. This is irritating as in existing literature this coefficient is described as the difference of thrust and power coefficient i.e.  $C_T - C_P$ . In contrast, I assume they mean a factor in front of difference  $C_T - C_P$ , resulting in  $C_{TKE}$  = anyFactor \* ( $C_T - C_P$ ), as it was pointed out in a recently published study (Archer et al. 2020). In case they really used a constant  $C_{TKE}$  for the comparison with the work of Siedersleben et al. (2020), the comparison is wrong.

Secondly, although they have shown that the bug corrected version of the wind farm parameterization of Fitch et al. (2012) has an large influence on the TKE within the rotor and wake area, the impact of the enhanced TKE on the wind speed is not discussed although this is the most crucial point of this study. I would assume that increased TKE would cause a weakening of the wakes due to enhanced mixing. This point should be added! A figure similar to Fig. 13 should be added for the horizontal wind field or Fig. 9a) should be replaced with the bug corrected Fitch parameterization.

Thirdly, a table with the conducted simulations would help the reader to have an overview about the different model setups.

**Specific comments:**

P1 L9: "However, their skill is limited... the farm edge." This is not related to the parameterizations, this is a purely numerical issue. Considered omitting this sentence.

P2 L18: The turbulence is modified by wind farms within the wake but not necessarily increased. For example, see Platis et al. (2018).

P3 L59: The Bundesnetzagentur does not host any open source flight data! The Bundesnetzagentur does host wind turbine location data!

P5 L128: Could you please comment why you used WRF 3.7 and not a newer version of WRF by now WRF-4.2.2 is already released? Is the EWP scheme only compatible with WRF 3.7?

P5 L136: The spin-up time is really short, especially with regards to your domain with an 18 km horizontal grid? Have you worse results using a longer one?

P5 L148: The  $C_{TKE}$  coefficient wasn't set to 1 in S2020! They used the default WFP in which  $C_{TKE} = C_T - C_P$ ! Please comment if you have modified the code within the Fitch scheme. If you have done so a comparison with the results of S2020 is difficult! Could it be that you mean:  $C_{TKE} = factor^* (C_T - C_P)$ ?

P5 L154: See comment before, in case you used the power coefficients and thrust coefficients and you haven't modified the code in WRF 3.7 in module\_fitch.F, your  $C_{TKE}$  is not constant i.e. unity. Please comment on that!

P6 L164: How have you determined the initial length scale for the EWP? Have you conducted a set of sensitivity studies to get a best fit? Was the same initial length scale used in Agora Energiewende et al. (2020)

P7 L192-204 Is this comparison really necessary, you are comparing a point measurement to a vertical profile?

**Technical comments:**

P2 L54: ... the increased wind resource  $\rightarrow$  increased wind resources?

P3 L79 ... here as Fig. 2 -> in Fig. 2?

P3 L87 Horizontal flight data  $\rightarrow$  horizontal flight data

P4 L124 ERA5: Be more precisely here, where did you get the data from? In case you have downloaded it from the Copernicus Climate Center you should definitely cite the source according to their terms of usage!

Fig. 2 The green and red colors are not color-blind safe, i.e. the flight track 1 and 3 are hardly to distinguish. Please consider redoing this figure!

Fig. 3 An indication of the location of the close-up shown in Fig. 3b) in the Fig 3a) would help the reader.

Fig. 5 These colors aren't the state of the art any more. Please consider redoing this figure as they are not colorblind safe. Is the TKE scale of 5e) similar to 5f). I am wondering as the last tick label in 5f) is missing.

Fig. 6 Could you please indicate the rotor-area, that would make it easier to follow the corresponding text.

Fig. 11 Rainbow is dead. Please consider redoing this figure using color-blind colorbars as pointed out in several publications (Stauffer et al. 2015; Thyng et al. 2016...).

Fig. 11 - 12: First of all: Please add a clear caption describing what is actually show in Fig. 12. It is possible to guess that (a-b) is showing wind speed due to the order of magnitude compared to (c-d). However, someone who is not that familiar with the units of TKE might have difficulties to draw the correct conclusions. Please use up to date colorbars as it is mentioned in the technical section (rainbow is dead).

Secondly, describe where this cross-section is located within the caption and make clear how this cross-section is orientated i.e. south-north or north-south!

**Code and data availability**

On 29 January 2021 the reviewer could not access the following address: https://zenodo.org/record/4133350.X5aZOO3cBaR

The authors should provide a corrected module\_pbl\_driver.F file, to account for the bug in wind farm parameterization of Fitch et al. (2020). In the official WRF repository they provide, the WRF bug is not fixed for version 3.7.

**Literature**

- Agora Energiewende, Technical University of Denmark, Max-Planck-Institute for Biogeochemistry, and Agora Verkehrswende, 2020: Making the Most of Offshore Wind: Re-Evaluating the Potential of Offshore Wind in the German North Sea.
- Archer, C. L., S. Wu, Y. Ma, and P. A. Jiménez, 2020: Two Corrections for Turbulent Kinetic Energy Generated by Wind Farms in the WRF Model. *Mon. Weather Rev.*, **148**, 4823– 4835, https://doi.org/10.1175/MWR-D-20-0097.1.
- Fitch, A. C., J. B. Olson, J. K. Lundquist, J. Dudhia, A. K. Gupta, J. Michalakes, and I. Barstad, 2012: Local and Mesoscale Impacts of Wind Farms as Parameterized in a Mesoscale NWP Model. *Mon. Weather Rev.*, **140**, 3017–3038, https://doi.org/10.1175/MWR-D-11-00352.1.
- Platis, A., and Coauthors, 2018: First in situ evidence of wakes in the far field behind offshore wind farms. *Sci. Rep.*, **8**, 2163, https://doi.org/10.1038/s41598-018-20389-y.
- Siedersleben, S. K., and Coauthors, 2020: Turbulent kinetic energy over large offshore wind farms observed and simulated by the mesoscale model WRF (3.8.1). *Geosci. Model Dev.*, **13**, 249–268, https://doi.org/10.5194/gmd-13-249-2020.
- Stauffer, R., G. J. Mayr, M. Dabernig, and A. Zeileis, 2015: Somewhere Over the Rainbow: How to Make Effective Use of Colors in Meteorological Visualizations. *Bull. Am. Meteorol. Soc.*, 96, 203–216, https://doi.org/10.1175/BAMS-D-13-00155.1.
- Thyng, K., C. Greene, R. Hetland, H. Zimmerle, and S. DiMarco, 2016: True Colors of Oceanography: Guidelines for Effective and Accurate Colormap Selection. *Oceanography*, **29**, 9–13, https://doi.org/10.5670/oceanog.2016.66.

Volker, P. J. H., J. Badger, A. N. Hahmann, and S. Ott, 2015: The Explicit Wake Parametrisation V1.0: a wind farm parametrisation in the mesoscale model WRF. *Geosci. Model Dev.*, **8**, 3715–3731, https://doi.org/10.5194/gmd-8-3715-2015.

---

## Referee Comment (RC2) · Anonymous Referee #2 · 23 Feb 2021

**Review of Larsén and Fischereit - gmd-2020-358**

General

The study by Larsén and Fischereit investigates two different wind farm wake parametrisation that are in principle available for the mesoscale model WRF during a situation with low level jets. Thus, the study generally addresses two important topics of mesoscale meteorology: the simulation of low level jets as well as the topic of larger scale wake effects and could thus in principle be an important scientific contribution.

However, there are several major points and a number of minor points that the authors should address before consideration the publication as research paper in GMD. As these points require from my point of view re-simulation and re-interpretation of parts of the results, I recommend publication after major revisions.

Major Points

1. **Bug in Fitch parametrization:** The authors mention the bug in the Fitch parametrization that was announced and corrected in June 2020 and discussed in the publication by Archer et al. 2020. However, large parts of the results that the author discuss are due to this bug in the parametrization. One prominent example is the high TKE above the farm, e.g. in Fig. 11c. The reason that other studies like the Siedersleben et al., 2020 are affected as well is not a reason for obviously using a parametrization that contains a bug! Thus, I see the need for correcting the bug in the WRF version used by the authors, re-running and re-discussing all results with this new version.

2. **Erroneous turbine data:** Table 2 contains several wrong information about the wind farm details. These should be corrected and the simulations re-run. The ones I could identify are: Alpha Ventus contains of 12 turbines of two different turbine types (Adwen M5000-116 (southern part) and Senvion 5M (northern part)), Bard Offshore contains of 80 turbines of the BARD 5.0 turbine with a rotor diameter of 122 m. Horns Rev 2 has a hub height of 68 meters. Please correct those and carefully check all others and re-run the model simulations.

3. **Reproducibility:** The authors are referring to the Volker et al, 2015 study for the availability of the EWP model. However that one refers to a zenodo record https://zenodo.org/record/33435 that contains the parametrization for the WRF

model version 3.4. The authors however have used WRF version 3.7. So, the actual EWP model code used in this study should be provided along with the study.

4. **Language:** In addition, the study would benefit from a thorough review of the language as it is hard to read in several parts. The authors for example make several times use of in-line enumerations which should either be indented or rewritten so that these sentences are easier to understand.

Minor Points

1. **P1-L15**: "For instance in the North Sea..." ⇒ I recommend introducing what a wind farm and a wind farm cluster is from your point of view. Also please set the reference to 4COffshore as a proper reference.

2. **P1-L21**: "most-used mesoscale model" ⇒ for the application of wind farm wakes

3. **P2-L24**: "the two most commonly applied explicit wind farm paramtrizations" ⇒ I suggest to explain in one sentence what the difference between an implicit and an explicit wake scheme are.

4. **P2-L50**: "... occurred on about 65 % of the days during the campaign." ⇒ This is true but it should be mentioned here that this is not 65 % of the time but could also be a short period of the day.

5. **P3-L59**: "... open source flight data from Bundesnetzagentur ..." ⇒ The Bundesnetzagentur which is the federal grid agency does not provide flight data. This is confusing here as the turbine coordinates originate from them not the flight data.

6. **P3-L77**: "is the open access measurements" ⇒ Grammar and language issue - Maybe: The first are the publicly available airborne measurement data?

7. **P3-L82**: "The flight data include (1) ..." ⇒ One of the enumerations mentioned in Major Point 4 that should be indented or reformulated.

8. **P3-L90**: "... corresponding to a horizontal resolution of 0.66 m ..." ⇒ How is this calculated? With he speed above ground of the aircraft?

9. **P4-L94**: "The choice of the window length ..." ⇒ I did not understand his sentence. Please revise.

10. **P4-L96**: "... order of a couple of minutes, which is a reasonable time scale" ⇒ Reasonable for what? Comparing to the models output/time step/horizontal averaging?

11. **P4-L101 and Fig. 3b**: Please use another color than the blue dots as the turbine positions are also blue.

12. **P4-L104**: "The second measurement type is from the ..." ⇒ Do you mean that the second dataset originates from the FINO1 met mast?

13. **P4-L105**: "FINO 1 is in the wake of the upstream wind farm Borkum Riffgrund..." ⇒ Again difficult to understand. Do you mean: In this situation, FINO1 is located in the wake of the wind farm Borkum Riffgrund that is operating XX km upstream?

14. **P4-L115**: "LLJs over the Southern North sea ... Tay et al., 2020)" ⇒ This whole paragraph should be shifted to the introduction.

15. **P4-L118**: "... WRF, where important elements include model domain configuration,..." ⇒ Important elements for what?

16. **P4-L122**: "This includes (1) ..." ⇒ see major comment 4.

17. **P5-L125**: "... others in Tay et al. (2020), while in Nunalee and Basu (2014), MYNN performed fine but best candidate was QNSE ..." ⇒ Are these the same

sites? Also, there are six years between these studies. Implementations might also change considerably over these years.

18. **P5-L128**: "We use WRF version 3.7 to simulate this case ... (Stark et al., 2007) was used." ⇒ Please consider putting this model setup into a tabular overview rather than running text.

19. **P5-L153**: "This problem is solved by increasing the land area in the southern part." ⇒ I guess you did not artificially increase the land but shifted ore increased the domain don't you?

20. **P5-158**: "... and manually corrected to fit the wind farm shapes from emodnet" ⇒ Good that you mention that you have corrected the coordinates. The Bundesnet-zagentur data are known to be erroneous. Sometimes turbines are also missing. Did you make sure that the correct number of turbines per farm are included?

21. **P5-L159**: "with the simulated date (Table 2)." ⇒ I guess you measurement time?

22. **P6-L175**: "jet nose, with the lowest ones beneath 200 m and the highest ones at 350-400 m, suggesting the presence of multiple internal boundary layers in associated with the flow from the land." ⇒ Is that really true or did the jet core move with height as there is considerable time between the measurement of the profiles?

23. **P6-L182**: "... from upstream Borkum Riffgrund wind farm" ⇒ Do you mean "orig-inating form the Borkum Riffgrund wind farm that is located upstream"?

24. **P6-L184**: "Six 10-min modeled data ..." ⇒ This sentence is very hard to under-stand. Please revise.

25. **P6-L186**: ⇒ One of the enumerations mentioned in Major Point 4 that should be indented or reformulated.

26. **P6-L188**: "Thus the average values from the surface to the rotor top height are comparable between the two schemes." ⇒ Please add "in this situation".

27. **P7-L197**: "The above descriptions of the wind speed for FINO1 are also true for point A, as can be seen in Fig. 7a. Here the EWP scheme provides a better estimate of mean wind speed." ⇒ You should also mention/discuss here which profile looks physically more sound. The EWP scheme just provides more shear and better values at measurement height but the Fitch one looks closer to measured and high-fidelity modeled wake profiles.

28. **P7-L200**: "TKE from the Fitch scheme increases significantly with height, and for the value at point A, it is overestimated in comparison with the flight data." ⇒ Is that still true without the bug in the parametrization?

29. **P7-L207**: "Here the modeled values at FINO 1 are weighted between two closest grid points (one inside and one outside the farm) according to the distances between the grid points and the mast location." ⇒ I am not convinced that it is physically meaningful at all to compare data from a model grid point where a parametrization is active to measurement data. Why didn't you just use the data from the first grid point upstream?

30. **P7-L218**: "Without taking wind farm wake into..." ⇒ Do you mean "the wind farm wakes"?

31. **P8-L233**: "This is a phenomenon that deserves further investigation (Djath et al., 2018) but is beyond the scope of this study." ⇒ Do you mean the flow below the rotor, which is in particular strong during stable stratification?

32. **P8-L256**: "The abrupt increase in the TKE in the same aera is likely related to this flow acceleration and is also missing in the WRF results." ⇒ Couldn't also the different jet core heights of simulations/measurements be a reason for this?

33. **Discussion/Conclusion**: The introduction of the discussion section has the character of a conclusion. I recommend combining and restructuring sections 4 and 5.

34. **P10-L320**: Low level jets and wind farm wakes have been investigated in numerous studies. In particular LES. Long-distance / mesoscale wakes might be true.

35. **P11-L329**: The zenodo link is not working.

36. **P11-L335**: Are FINO1 station data really available from the PANGEA database?

37. **P12 - References**: Several References contain several URLs, some URLs are in italic font.

38. **P12 - L349**: "Bärfuss K. ..." ⇒ That reference looks strange.

39. **Figure 9,11,12,13**: Please add the quantity and unit shown to the color bars. They are provided in the captions only.

---

## Author Response (AR1)

**Response to the comments about the submitted paper**

**A case study of wind farm effects using two wake parameterizations in WRF (V3.7.1) in the presence of low level jets**

We thank the reviewers for his constructive comments. We have addressed each and every one of them and modified the paper accordingly. Our detailed answers follow.

Please note that reviewers' comments are in italics while our answers are not. Additions to the original manuscript are indicated in blue.

There are three parts in this file: Answers to reviewer 1, answers to reviewer 2 and attachment README.md.

Please read the README.md-file at the end this file for all the updates.

**Answers to Reviewer 1**

**General comment:**

*In general, this work makes an important contribution to the wind energy resource assessment. The two most prominent wind farm parametrizations are evaluated by the use of airborne measurements and are compared against each other. This is important as the wind farm parameterization of Volker et al. (2015) has been used in a recently published report, indicating that wakes in the German Bight could have a massive impact on the energy harvesting in the future, although this parameterization was so far only poorly evaluated for offshore wind farms. However, three major points have to revised in this manuscript:*

**Comment R1.M1** *First of all, the description of the TKE coefficient (CTKE) that determines the amount of TKE added by the wind farm parameterization is highly confusing. In this manuscript they authors describe CTKE as a constant value. This is irritating as in existing literature this coefficient is described as the difference of thrust and power coefficient i.e. CT - CP. In contrast, I assume they mean a factor in front of difference CT - CP, resulting in CTKE = anyFactor \* (CT - C P), as it was pointed out in a recently published study (Archer et al. 2020). In case they really used a constant CTKE for the comparison with the work of Siedersleben et al. (2020), the comparison is wrong.*

**Answer to R1.M1** We thank the reviewer for pointing this out. Indeed it is confusing that we have used $C_{TKE}$ for the correction factor. In fact it is as the reviewer speculated that it is the correction factor we are addressing to the numbers 1 and 0.25. In the new text we re-wrote it, so that $C_{TKE} = \alpha(C_T - C_P)$, and $\alpha$ is used both in the main text and the figure captions.

**Comment R1.M2** *Secondly, although they have shown that the bug corrected version of the wind farm parameterization of Fitch et al. (2012) has an large influence on the TKE within the rotor and wake area, the impact of the enhanced TKE on the wind speed is not discussed although this is the most crucial point of this study. I would assume that increased TKE would cause a weakening of the wakes due to enhanced mixing. This point should be added! A figure similar to Fig. 13 should be added for the horizontal wind field or Fig. 9a) should be replaced with the bug corrected Fitch parameterization.*

**Answer to R1.M2** It is a good point. Results and analysis from the Fitch scheme with advection on are included (see new figures from Fig. 6 to 15). This includes the corresponding spatial distribution of wind speed as Fig. 14 as suggested by the reviewer.

**Comment R1.M3** *Thirdly, a table with the conducted simulations would help the reader to have an overview about the different model setups.*

**Answer to R1.M3** Good point. Such a table is now made available in the new vision as Table 3.

**Specific comments:**

**Comment R1.1** *P1 L9: "However, their skill is limited... the farm edge." This is not related to the parameterizations, this is a purely numerical issue. Considered omitting this sentence.*

**Answer to R1.1** We agree that in order to reach the conclusion as we gave, more work needs to be done. This sentence is therefore removed.

**Comment R1.2** *P1 L18: The turbulence is modified by wind farms within the wake but not necessarily increased. For example, see Platis et al. (2018).*

**Answer to R1.2** We agree that the response of turbulence with changing wind conditions could be rather complicated. Under wake effect, turbulence is enhanced comparing to same wind speed, but it can also decrease with reduced wind speed, resulting in a net value that is not necessarily higher than a condition in the absence of wind farm. To avoid confusion, we changed "increase" to "change".

**Comment R1.3** *P3 L59: The Bundesnetzagentur does not host any open source flight data! The Bundesnetzagentur does host wind turbine location data!*

**Answer to R1.3** Indeed. This reference was added by mistake previously and is now removed. Thank you for pointing this out.

**Comment R1.4** *P5 L128: Could you please comment why you used WRF 3.7 and not a newer version of WRF by now WRF-4.2.2 is already released? Is the EWP scheme only compatible with WRF 3.7?*

**Answer to R1.4** It is of course interesting to find out how each version of WRF produces results and if the latest version is the best. Though, regarding the purpose of this study, the importance of the version investigation is secondary. We expect the main findings be consistent with different versions of WRF. Also, EWP is compatible with any version of WRF.

**Comment R1.5** *P5 L136: The spin-up time is really short, especially with regards to your domain with an 18 km horizontal grid? Have you worse results using a longer one?*

**Answer to R1.5** It is a relevant topic: how long is long enough for a spin-up time. A long spin-up time is essential when we would like to resolve the more or less stationary mesoscale variabilities. At other times, when a special atmospheric phenomenon is developing, it is more important to make sure that the initial conditions are properly introduced to the simulation: in this case, the development of low level jet. We see no obvious problems in running the simulation this way when most of the analysis are in the second half of the simulation.

**Comment R1.6** *P5 L148: The CTKE coefficient wasn't set to 1 in S2020! They used the default WFP in which CTKE = CT – CP! Please comment if you have modified the code within the Fitch scheme. If you have done so a comparison with the results of S2020 is difficult! Could it be that you mean: CTKE = factor\* (CT – CP)?*

**Answer to R1.6** Indeed as the reviewer says. We made the correction. See our response in answer to comment R1.2.

**Comment R1.7** *P5 L154: See comment before, in case you used the power coefficients and thrust coefficients and you haven't modified the code in WRF 3.7 in module_fitch.F, your CTKE is not constant i.e. unity. Please comment on that!*

**Answer to R1.7** See our response in answer to comment R1.2.

**Comment R1.8** *P6 L164: How have you determined the initial length scale for the EWP? Have you conducted a set of sensitivity studies to get a best fit? Was the same initial length scale used in Agora Energiewende et al. (2020)*

**Answer to R1.8** We agree this sentence is oversimplified and it has been re-written. In the

literatures, values 1.5 (e.g. Agora Energiewende et al. (2020)) and 1.7 (Volker et al. 2015) have been used. In Volker et al. (2015) it was also shown that the model output has only negligible difference for values between 1.5 to 1.9 that are used. For our study, we tested the values from 1.5 to 1.7 and it showed almost no difference. Thus the value 1.6 was used.

**Comment R1.9** *P7 L192-204 Is this comparison really necessary, you are comparing a point measurement to a vertical profile?*
**Answer to R1.9** The reviewer is right on this. We removed this comparison in the new version.

**Technical comments:**

**Comment R1.10** *P2 L54: ... the increased wind resource → increased wind resources?*
**Answer to R1.10** Suggestion taken.

**Comment R1.11** *P3 L79 ... here as Fig. 2 → in Fig. 2?*
**Answer to R1.11** Suggestion taken

**Comment R1.12** *P3 L87 Horizontal flight data → horizontal flight data*
**Answer to R1.12** Suggestion taken

**Comment R1.13** *P4 L124 ERA5: Be more precisely here, where did you get the data from? In case you have downloaded it from the Copernicus Climate Center you should definitely cite the source according to their terms of usage!*
**Answer to R1.13** The source of the data has been provided in the data-availability section and a citation is added.

**Comment R1.14** *Fig. 2 The green and red colors are not color-blind safe, i.e. the flight track 1 and 3 are hardly to distinguish. Please consider redoing this figure!*
**Answer to R1.14** Thanks for pointing this out, which we haven't given it a good thought before. We have revised all figures using color-blind safe color-codes and checked that through the firefox add-on. For figure 2 we used colors based on the 'Color Cycle Picker' `https://github.com/mpetroff/color-cycle-picker`.

**Comment R1.15** *Fig. 3 An indication of the location of the close-up shown in Fig. 3b) in the Fig 3a) would help the reader.*
**Answer to R1.15** Suggestion taken

**Comment R1.16** *Fig. 5 These colors aren't the state of the art any more. Please consider redoing this figure as they are not colorblind safe. Is the TKE scale of 5e) similar to 5f). I am wondering as the last tick label in 5f) is missing.*
**Answer to R1.16** The figures are revised as suggested. The same color-code as in figure 2 has been used.

**Comment R1.17** *Fig. 6 Could you please indicate the rotor-area, that would make it easier to follow the corresponding text.*

**Answer to R1.17** Suggestion taken.

**Comment R1.18** *Fig. 11 Rainbow is dead. Please consider redoing this figure using color-blind colorbars as pointed out in several publications (Stauffer et al. 2015; Thyng et al. 2016...).*
**Answer to R1.18** We used color-blind safe colormap viridis is used in the new version.

**Comment R1.19** *Fig. 11 - 12: First of all: Please add a clear caption describing what is actually show in Fig. 12. It is possible to guess that (a-b) is showing wind speed due to the order of magnitude compared to (c-d). However, someone who is not that familiar with the units of TKE might have difficulties to draw the correct conclusions. Please use up to date colorbars as it is mentioned in the technical section (rainbow is dead). Secondly, describe where this cross-section is located within the caption and make clear how this cross-section is orientated i.e. south-north or north-south!*
**Answer to R1.19** Suggestions taken. The south-north coordination is expected to be seen as the latitude increases on the x-axes.

**Code and data availability**

**Comment R1.20** *On 29 January 2021 the reviewer could not access the following address:* `https: // zenodo. org/ record/ 4133350. X5aZOO3cBaR`
**Answer to R1.20** Apologies for that! The link was `https://zenodo.org/record/4133350#` `.X5aZOO3cBaR`. Along with the review, we updated the link corresponding to the new version and the link is: https://doi.org/10.5281/zenodo.4668613
.

**Comment R1.21** *The authors should provide a corrected module_pbl_driver.F file, to account for the bug in wind farm parameterization of Fitch et al. (2020). In the official WRF repository they provide, the WRF bug is not fixed for version 3.7.*
**Answer to R1.21** The file along with a detailed description in the README is provided in the new version of our zenodo-repository: https://doi.org/10.5281/zenodo.4668613. See in the directory $\text{Modified}_W RF/Phys$

**Answers to Reviewer 2**

**General**

*The study by Larsén and Fischereit investigates two different wind farm wake parametrisation that are in principle available for the mesoscale model WRF during a situation with low level jets. Thus, the study generally addresses two important topics of mesoscale meteorology: the simulation of low level jets as well as the topic of larger scale wake effects and could thus in principle be an important scientific contribution. However, there are several major points and a number of minor points that the authors should address before consideration the publication as research paper in GMD. As these points require from my point of view re-simulation and re-interpretation of parts of the results, I recommend publication after major revisions.*

**Major Points**

**Comment R2.M1**  *Bug in Fitch parametrization: The authors mention the bug in the Fitch parametrization that was announced and corrected in June 2020 and discussed in the publication by Archer et al. 2020. However, large parts of the results that the author discuss are due to this bug in the parametrization. One prominent example is the high TKE above the farm, e.g. in Fig. 11c. The reason that other studies like the Siedersleben et al., 2020 are affected as well is not a reason for obviously using a parametrization that contains a bug! Thus, I see the need for correcting the bug in the WRF version used by the authors, re-running and re-discussing all results with this new version.*

**Answer to R2.M1** This is a good point. The previous version of text may not have made these following points clear. The bug is that the turbine-induced TKE is not corrected implemented in the TKE advection scheme. Siedersleben et al., 2020 used the version with the bug included, and they came to the conclusion that "with advection turned-off gives better results". In our previous version, we actually used the bug-corrected version, but also with advection turned-off. In the new version, most analyses are based on the bug-corrected version, including both advection-on and advection-off. New figures are made (Fig.6 to Fig. 15). A brief discussion is also provided on the differences brought by the bug, in order to give an idea how much one can refer to existing publications that are based on the bug version.

**Comment R2.M2**  *Erroneous turbine data: Table 2 contains several wrong information about the wind farm details. These should be corrected and the simulations re-run. The ones I could identify are: Alpha Ventus contains of 12 turbines of two differ- ent turbine types (Adwen M5000-116 (southern part) and Senvion 5M (northern part)), Bard Offshore contains of 80 turbines of the BARD 5.0 turbine with a rotor diameter of 122 m. Horns Rev 2 has a hub height of 68 meters. Please correct those and carefully check all others and re-run the model simulations.*

**Answer to R2.M2** Thank you for spotting this. There was indeed a mistake in the Table regarding the Horns Rev 2 hub height, though we confirm that we used 68 m as hub height in the actual simulation – it was a typo in the Table. We also checked all the other turbine models and corrected any other mistakes. Please also note that we did not write clearly that this table does not contain the actual wind turbine models for each farm, but the turbine models that we used for the different farms. We tried to use the actual turbine model as far as possible, but unfortunately thrust and power curves were not available to us for all turbine models. In those cases we used

a replacement model. Details on these replacements have been added to our zenodo-README and it is now more clearly stated in the text as well as in the table caption. We apologize for any confusion.

We also report here that by correcting these turbines (hub height in Gode Wind, different turbine models in alpha ventus), the results are very similar, suggesting these turbines do not have a large effect on the existing analysis in this study, with Fig. R1 as an example.

[Figure]

Figure R1: As in Fig.7 in the main text, comparing wind profiles before and after the correction of wind turbines (solid vs dotted lines) for $\alpha = 1$ and 0.25, respectively.

**Comment R2.M3** *Reproducibility: The authors are referring to the Volker et al, 2015 study for the availability of the EWP model. However that one refers to a zenodo record* `https://zenodo.org/record/33435` *that contains the parametrization for the WRF model version 3.4. The authors however have used WRF version 3.7. So, the actual EWP model code used in this study should be provided along with the study.*

**Answer to R2.M3** It is a good point. While the physics behind the EWP-scheme has not changed much since the original publications in 2015, the structure of the routine has. We also update that we actually used WRF version 3.7.1. The source code to the EWP scheme (file `module_wf_ewp.F`), along with other files that need to be modified (10 in total) are provided based on WRF version 3.7.1 on the zenodo-repository, now. In the README, we describe in detail, how to reproduce the simulations (starting from cloning the official WRF release, checking out v3.7.1 and add the changes). In a separate sub-directory in the zenodo-repository, we also uploaded the modified files for using the bug-fixed Fitch scheme, along with installation instructions in the README. Please read the attached README.md file for the updates. We understood it that we will update the new version of our zenodo-repository as the next step action, following this response-to-reviewer.

**Comment R2.M4** *Language: In addition, the study would benefit from a thorough review of the language as it is hard to read in several parts. The authors for example make several times use of in-line enumerations which should either be indented or rewritten so that these sentences are easier to understand.*

**Answer to R2.M4** We went through the language several time again and paid special attention to the use of in-line enumerations.

**Minor Points**

**Comment R2.1** *P1-L15: "For instance in the North Sea..." → I recommend introducing what a wind farm and a wind farm cluster is from your point of view. Also please set the reference to 4COffshore as a proper reference.*
**Answer to R2.1** The sentence regarding a wind farm and wind farm cluster has been re-written. The reference to 4Coffshore is provided.

**Comment R2.2** *P1-L21: "most-used mesoscale model" → for the application of wind farm wakes*
**Answer to R2.2** The sentence is revised so that this message is clear.

**Comment R2.3** *P2-L24: "the two most commonly applied explicit wind farm paramtrizations" → I suggest to explain in one sentence what the difference between an implicit and an explicit wake scheme are.*
**Answer to R2.3** Suggestion taken. We added this sentence: "There are mainly two kinds of methods to parameterize wind farm effect, one is the implicit method that parameterize the effect through an increase in surface roughness length and the other one is the explicit method that parameterize the effect through elevated momentum sink."

**Comment R2.4** *P2-L50: "... occurred on about 65 % of the days during the campaign." → This is true but it should be mentioned here that this is not 65 % of the time but could also be a short period of the day.*
**Answer to R2.4** The sentence has been revised as "...they found that on about 65% of the days during the campaign LLJs occurred part time of the days."

**Comment R2.5** *P3-L59: "... open source flight data from Bundesnetzagentur ..." → The Bundesnetzagentur which is the federal grid agency does not provide flight data. This is confusing here as the turbine coordinates originate from them not the flight data.*
**Answer to R2.5** Yes, indeed. This reference was added previously by mistake. It has been removed in the new version.

**Comment R2.6** *P3-L77: "is the open access measurements" → Grammar and language issue - Maybe: The first are the publicly available airborne measurement data?*
**Answer to R2.6** The sentence is re-written.

**Comment R2.7** *P3-L82: "The flight data include (1) ..." → One of the enumerations mentioned in Major Point 4 that should be indented or reformulated.*
**Answer to R2.7** Suggestion taken. We removed the enumerations.

**Comment R2.8** *P3-L90: "... corresponding to a horizontal resolution of 0.66 m ..." → How is this calculated? With he speed above ground of the aircraft?*
**Answer to R2.8** Yes, with the speed above ground of the aircraft. This sentence has been re-written.

**Comment R2.9** *P4-L94: "The choice of the window length ..." → I did not understand his sentence. Please revise.*

**Answer to R2.9** We replaced "window length" with "data length", in order to avoid confusion and revised the whole sentence.

**Comment R2.10** *P4-L96: "... order of a couple of minutes, which is a reasonable time scale" → Reasonable for what? Comparing to the models output/time step/horizontal averaging?*

**Answer to R2.10** This sentence has been re-written, and now it reads "...on the order of a couple of minutes, which is a reasonable integral time scale for separating boundary-layer turbulence and external fluctuations"

**Comment R2.11** *P4-L101 and Fig. 3b: Please use another color than the blue dots as the turbine positions are also blue.*

**Answer to R2.11** Suggestion taken, blue is changed to purple.

**Comment R2.12** *P4-L104: "The second measurement type is from the ..." → Do you mean that the second dataset originates from the FINO1 met mast?*

**Answer to R2.12** Yes. The sentence has been re-written.

**Comment R2.13** *P4-L105: "FINO 1 is in the wake of the upstream wind farm Borkum Riffgrund..." → Again difficult to understand. Do you mean: In this situation, FINO1 is located in the wake of the wind farm Borkum Riffgrund that is operating XX km upstream?*

**Answer to R2.13** The sentence has been re-written as "a wind direction from about 240° (Fig. 5), the flow passes the wind farm Borkum Riffgrund before reaching $F1$, resulting in reduced wind speed downwind of the farm, including at FINO 1 (see Fig. 1a)"

**Comment R2.14** *P4-L115: "LLJs over the Southern North sea ... Tay et al., 2020)" → This whole paragraph should be shifted to the introduction.*

**Answer to R2.14** We moved this part on the physics of LLJ to the introduction (page 2, line 53-55), but keep the part of discussion for model setup.

**Comment R2.15** *P4-L118: "... WRF, where important elements include model domain configuration,..." → Important elements for what?*

**Answer to R2.15** This sentence has been re-written to"Important elements for accurately simulating LLJs using WRF include model domain configuration, initialization and boundary forcing data, horizontal and vertical spatial resolutions, PBL schemes etc., as have been explored in e.g. Nunalee and Basu (2014); Wagner et al. (2019); Kalverla et al. (2019); Siedersleben et al. (2020); Tay et al. (2020). "

**Comment R2.16** *P4-L122: "This includes (1) ..." → see major comment 4.*

**Answer to R2.16** This paragraph has been re-written according to the reviewer's suggestion.

**Comment R2.17** *P5-L125: "... others in Tay et al. (2020), while in Nunalee and Basu (2014), MYNN performed fine but best candidate was QNSE ..." → Are these the same sites? Also, there are six years between these studies. Implementations might also change considerably over these years.*

**Answer to R2.17** No, these are not the same sites. The text is revised to make it clear that we choose MYNN since it is the only one that works with the Fitch scheme.

**Comment R2.18** *P5-L128: "We use WRF version 3.7 to simulate this case … (Stark et al., 2007) was used." → Please consider putting this model setup into a tabular overview rather than running text.*
**Answer to R2.18** Table 2 is created following the reviewer's suggestion.

**Comment R2.19** *P5-L153: "This problem is solved by increasing the land area in the southern part." → I guess you did not artificially increase the land but shifted ore increased the domain don't you?*
**Answer to R2.19** This sentence has been re-written: "This problem is solved by including more area in the model domain"

**Comment R2.20** *P5-158: "… and manually corrected to fit the wind farm shapes from emodnet" → Good that you mention that you have corrected the coordinates. The Bundesnetzagentur data are known to be erroneous. Sometimes turbines are also missing. Did you make sure that the correct number of turbines per farm are included?*
**Answer to R2.20** The turbine numbers are provided in Table 4 and double checked with different data sources. The updated table shows the turbine models used for the new simulations.

**Comment R2.21** *P5-L159: "with the simulated date (Table 2)." → I guess you measurement time?*
**Answer to R2.21** Yes we mean both the measurement time and the current studied case. The sentence has been revised.

**Comment R2.22** *P6-L175: "jet nose, with the lowest ones beneath 200 m and the highest ones at 350-400 m, suggesting the presence of multiple internal boundary layers in associated with the flow from the land." → Is that really true or did the jet core move with height as there is considerable time between the measurement of the profiles?*
**Answer to R2.22** Good point. Indeed, data from each profile were collected during a period of several minutes. During this period, the airplane has changed position significantly along the flight track. The multiple jet noses reflect therefore the non-homogeneity of flow in time and space. The corresponding text is revised accordingly.

**Comment R2.23** *P6-L182: "… from upstream Borkum Riffgrund wind farm" → Do you mean "originating form the Borkum Riffgrund wind farm that is located upstream"?*
**Answer to R2.23** Yes. The sentence has been re-written.

**Comment R2.24** *P6-L184: "Six 10-min modeled data …" → This sentence is very hard to understand. Please revise.*
**Answer to R2.24** Suggestion taken and the sentence has been revised.

**Comment R2.25** *P6-L186: → One of the enumerations mentioned in Major Point 4 that should be indented or reformulated.*
**Answer to R2.25** Suggestion taken and the paragraph has been revised.

**Comment R2.26**  *P6-L188: "Thus the average values from the surface to the rotor top height are comparable between the two schemes." → Please add "in this situation".*
**Answer to R2.26** Suggestion taken.

**Comment R2.27**  *P7-L197: "The above descriptions of the wind speed for FINO1 are also true for point A, as can be seen in Fig. 7a. Here the EWP scheme provides a better estimate of mean wind speed." → You should also mention/discuss here which profile looks physically more sound. The EWP scheme just provides more shear and better values at measurement height but the Fitch one looks closer to measured and high-fidelity modeled wake profiles.*
**Answer to R2.27** The corresponding sentence has been re-written. Though the authors are not clear what the reviewer meant by "high-fidelity modeled wake profiles"

**Comment R2.28**  *P7-L200: "TKE from the Fitch scheme increases significantly with height, and for the value at point A, it is overestimated in comparison with the flight data." → Is that still true without the bug in the parametrization?*
**Answer to R2.28** As suggested by the other reviewer, we removed the star (one point from the flight track) from the analysis, following the argument that it is merely one "snapshot" at one point compared to other analysis variation in time or space is included. But to answer the question of the reviewer, our results from the previous version are using the bug-corrected version, but with advection turned-off following S2020. Compared to advection turned-off, the values are only marginally smaller using a correction factor 1, but significantly smaller using a correction factor 0.25. But as discussed in the text, a correction factor of 0.25 does not necessarily bring improved results (e.g. Fig. 6, 8, 11). As we also pointed it out, neither Archer et al. (2020) nor the current study has sufficient measurements to decide which value of the correction factor is most suitable. In the new version, we added a few lines in the discussion section addressing the difference between the versions with and without the bug (page 11, line 332-337, line 349-354) .

**Comment R2.29**  *P7-L207: "Here the modeled values at FINO 1 are weighted between two closest grid points (one inside and one outside the farm) according to the distances between the grid points and the mast location." → I am not convinced that it is physically meaningful at all to compare data from a model grid point where a parametrization is active to measurement data. Why didn't you just use the data from the first grid point upstream?*
**Answer to R2.29** We also considered this option and made the choice of a weighted value eventually. In reality, FINO 1 is both under the wake impact of the upwind farm, and the blockage effect of the downwind farm. The horizontal wind speed gradient from the closest upwind grid point to FINO 1, which is over 1 km, is not reasonable to be ignored.

**Comment R2.30**  *P7-L218: "Without taking wind farm wake into..." → Do you mean "the wind farm wakes"?*
**Answer to R2.30** Yes. The sentence is revised.

**Comment R2.31**  *P8-L233: "This is a phenomenon that deserves further investigation (Djath et al.), but is beyond the scope of this study." → Do you mean the flow below the rotor, which is in particular strong during stable stratification?*
**Answer to R2.31** No. We meant that "in a short distance downwind of the wind farms, the surface wind at 10 m from the Fitch scheme suggests a slight speedup, see the brighter color in the farm wake shadows in Fig. 1a and the white color in Fig. 9b". We revised the sentence to make

this clear.

**Comment R2.32** *P8-L256: "The abrupt increase in the TKE in the same aera is likely related to this flow acceleration and is also missing in the WRF results." → Couldn't also the different jet core heights of simulations/measurements be a reason for this?*
**Answer to R2.32** It could be and it should worth exploring, e.g. with more measurements and high fidelity models. With data we have, it is difficult to draw further conclusions on this.

**Comment R2.33** *Discussion/Conclusion: The introduction of the discussion section has the character of a conclusion. I recommend combining and restructuring sections*
**Answer to R2.33** Agree. We re-structured and revised significantly the two sections.

**Comment R2.34** *P10-L320: Low level jets and wind farm wakes have been investigated in numerous studies. In particular LES. Long-distance / mesoscale wakes might be true.*
**Answer to R2.34** We revised this sentence.

**Comment R2.35** *P11-L329: The zenodo link is not working.*
**Answer to R2.35** Apologies. For the new version corresponding to the reply, the new link is https://doi.org/10.5281/zenodo.4668613

**Comment R2.36** *P11-L335: Are FINO1 station data really available from the PANGEA database?*

**Answer to R2.36** FINO 1 data are obtained from the Federal Maritime and Hydrographic Agency (BSH). This sentence has been revised.

**Comment R2.37** *P12 - References: Several References contain several URLs, some URLs are in italic font.*
**Answer to R2.37** Apologies for that. The reference list has been checked

**Comment R2.38** *P12 - L349: "Bärfuss K. ..." → That reference looks strange.*
**Answer to R2.38** The reference is corrected

**Comment R2.39** *Figure 9,11,12,13: Please add the quantity and unit shown to the color bars. They are provided in the captions only.*
**Answer to R2.39** The quantity unit are now shown to the color bars.

**Attachments:README.md**

**1 Documentation of LLJ WFP archive**

This archive documents the steps to reproduce the simulations in Larsén and Fischereit (2021).

**1.1 WPS steps**

- The WPS-namelist is included as `namelist.wps`
- The geogrid files for the WRF simulations are located in `geogrid/`
- Tables for running metgrid, ungrib and REAL are located in `tables/`
- The required ERA5 data can be downloaded using `get_era5_check_first.py` and the modified Vtable for ERA5 data is included in `tables/`

**1.2 WRF simulations**

Namelists and turbine-files to reproduce the simulations with EWP (`EWP_simulation/`), Fitch (`Fitch_simulation/`) and No wind farm simulation (`NWF_simulation/`) are included in the respective sub-directories.

**1.3 Mapping EWP and Fitch turbine files**

| Fitch turbine | EWP turbine | Info |
|---|---|---|
| wind-turbine-1.tbl | SWT-4.0-130 | |
| wind-turbine-2.tbl | SWT-3.6-120 | |
| wind-turbine-3.tbl | V80-2.0 | |
| wind-turbine-4.tbl | SWT-2.3-93 | |
| wind-turbine-6.tbl | M5000-116 | |
| wind-turbine-7.tbl | SWT-6.0-154 | 106 m hub height |
| wind-turbine-8.tbl | SWT-4.0-120 | |
| wind-turbine-9.tbl | 6.2M126_84 | All 6.2M126 models have identical thrust and power curves |
| wind-turbine-10.tbl | 6.2M126_95 | All 6.2M126 models have identical thrust and power curves |
| wind-turbine-16.tbl | 6.2M126_90 | All 6.2M126 models have identical thrust and power curves |
| wind-turbine-19.tbl | SWT-6.0-154_110 | same thrust and power curve as SWT-6.0-154 |
| wind-turbine-20.tbl | Senvion_5M | same thrust and power curve as M5000-116 |

The locations of the turbines are included in
`EWP_simulation/all_farms_corr_2017_new_LLJ_corrected.real` and
`Fitch_simulation/windturbines.txt`, respectively. The turbine characteristics have been extracted from https://www.wasp.dk/dataandtools#power-curves

**1.4 Fitch simulation with TKE advection bugfix**

In addition to a simulation with TKE advection off (`Fitch_simulation/namelist.input.Fitch`), two additional simulations have been conducted with TKE advection on and including the bugfix for the bug reported in Archer et al. (2020) (`Fitch_simulation/namelist.input.Fitch_adv_on_ctke_1` and `Fitch_simulation/namelist.input.Fitch_adv_on_ctke_025`). They differ in the used value for CTKE, which is a new wrf-namelist variable introduced by Archer et al. (2020). To conduct

the simulations, the WRF v3.7.1 source code was modified according to https://github.com/wrf-model/WRF/pull/1235 for those two simulations.

To include the bugfix the following steps have been performed:
1. git clone https://github.com/wrf-model/WRF.git
2. git checkout V3.7.1
3. git cherry-pick 962288fe (and resolve merge conflict for run/README.namelist)

For convienence the changed files are provided along the EWP source code in `modified_WRF/`

**1.5 Modified WRF v3.7.1**

The code changes made to WRF 3.7.1 are included in the directory `modified_WRF/`. The structure in this directory is the same as in the normal WRF 3.7.1. To reproduce the simulations modified files have been added in the respective sub-directories.

To reproduce the simulations use
1. git clone https://github.com/wrf-model/WRF.git
2. git checkout V3.7.1
3. add the files below (files with `..._EWP_FITCH_bugfixed` contain EWP and Fitch bugfix, whereas files without that addition will provide the Fitch version including the bugfix)

For EWP in total 10 changes/additions are required:
- 3 new files: `phys/module_wf_ewp.F` (core EWP module), `README.EWP` (README for this version of EWP) and `Registry/Registry.turbine` (EWP-specific namelist options)
- In `phys/module_physics_init.F` EWP is initialised alongside wind_fitch
- In `phys/module_pbl_driver.F` is called alongside wind_fitch
- In `phys/Makefile module_wf_ewp.F` has been added for compilation
- In `dyn_em/module_first_rk_step_part1.F` ROfrac has to be added to the argument list
- Registry modifications include `Registry/Registry.EM`, which needs to include `Registry.turbine` and `Registry/Registry.EM`, which sets ewpscheme as windfarm_opt=2
- In `main/depend.common` dependencies for `module_wf_ewp.F` are listed

For the TKE advection bugfix for the Fitch scheme:
- 5 files are changed in accordance with https://github.com/wrf-model/WRF/pull/1235 These are called `..._EWP_FITCH_bugfixed`

**1.6 PostProcessing of WRF outputs**

In the directory post_processing are: a mathematica script and two sub-folders [Toshare*MATLAB*scripts] and [Toshare*MATLAB*data]. Once the WRF simulations are done, using this mathematica script, one can extract data needed for relevant analysis and for making the figures in the paper. The script shows what data are extracted, how they are extracted and saved.

Data that are prepared from the mathematica script are stored in the directory [Toshare*MATLAB*data]. The directory [Toshare*MATLAB*scripts] includes a number of matlab scripts which use the data from [Toshare*MATLAB*data] and plots figures.

To reproduce Figure 1 of the paper, use `process_sar_paper.py`

**1.7 References**

Archer, C. L., Wu, S., and Ma, Y.: Two corrections for turbulent kinetic energy generated by wind farms in the WRF model, Monthly WeatherReview, DOI 10.1175/MWR-D-20-0097.1, 2020.

---

## Author Response (AR2)

The authors are grateful for the detailed suggestions from the reviewer. We give our response to these comments in the following, point-by-point. The reviewer's comments and suggestions are in blue and our responses are in black.

Congrats, the manuscript improved significantly. I have the following minor comments left:

Line 8: This is done using WRF with the code bug fixed …
-> This sounds strange: Maybe: This is done using a bug-fixed WRF version that includes the correct TKE advection following Archer et al. (2020).

*Reply*: Agreed. Suggestion taken.

Line 58: Wagner et al. (2019) showed that LLJs are a common phenomenon in …
-> The sentence is confusing. I suggest the following (please check if this is in-line with the findings of Wagner et al.): Wagner et al. (2019) showed that LLJs are a common phenomenon in the Southern North Sea. By analyzing 1.5 years of lidar and passive microwave radiometer data, they found that LLJ occurred on 65% of the days at least for a short period.

*Reply*: The original sentence from Wagner et al is "LLJs occurred at 14.5% of the time (449 of 3107 measurements) and on 64.8% (162 of 250) of the days". In this revised version we simply cited it the way it is, in order to avoid misunderstanding due to translation.

Line 144: The studies…
-> Better: These studies…
*Reply*: Suggestion taken.

Line 212-214: For some wind farms, e.g. Alpha Ventus …
-> I think it would be good to mention all turbine types, where the actual turbine data was not used but a similar model was used.
*Reply*: Good suggestion. The corresponding text has been re-written to "For Alpha Ventus and BARD Offshore, we could not obtain the thrust and power coefficients for the actual turbine. Therefore, the power and thrust curves of M5000-116 were scaled from the NREL 5 MW turbine. The Senvion 6.2M126 turbine in Nordsee One, OWP Nordergründe and OWP Nordsee Ost was similarly scaled from the DTU 10 MW reference turbine. Other power and thrust curves have been taken from Langor (2019) or from WAsP (http://www.wasp.dk/). In Table 2 the wind farms are listed with the turbine model used in the simulations". In addition, in Table 2, these scaled turbines are marked and explained in the Table caption.

Line 440: It remains inconclusive… (2 times)
-> I understand these sentences but I think both sentences would benefit from some elaboration. Please briefly explain "why".

*Reply:* Agreed. The corresponding text has been revised for both sentences. The new text reads: "It remains inconclusive which scheme is better at describing the wind field, as sometimes the EWP scheme outperforms the Fitch scheme, and some other times, it is the other way around. It also remains inconclusive which correction factor should be used in connection with the turbine-induced TKE generation in the Fitch scheme: we only tested two factors (1 and 0.25) here and we observe a better performance when using alpha=1 than alpha=0.25, which does not support the conclusion from Archer et al. (2020)".

Line 443: Neither scheme can not capture…
-> I guess you mean "Neither scheme can capture…"
*Reply:* indeed, it is now corrected. Thanks for spotting this.

Line 307: This is also often called …
-> I suggest to "this is referred to global blockage effect" and a recently discussed topic. And please refer to one or 2 recent relevant studies, e.g. https://www.mdpi.com/1996-1073/11/6/1609 or https://wes.copernicus.org/articles/6/521/2021/

*Reply*: Suggestions taken. The text was revised accordingly and the two studies were referred to.

Line 355: These characteristics can also be seen in a bird view of the spatial…
-> I suggest to use "horizontal wind field at XXXm height" instead of bird view.
*Reply*: suggestion taken.

Table 2: To my knowledge, the official reference of Ostia is:
https://www.sciencedirect.com/science/article/pii/S0034425711002197
*Reply*: Thanks for pointing this out. The reference has been corrected.

Everywhere: Please use the abbreviation Sect. instead of Sec. in accordance with Journal guidelines

*Reply*: Suggestion taken.

---

## Author Response (AR3)

2021-05-13

Dear Editor

The authors are pleased with the editor's decision that our paper is accepted for publication.

Though on "my author overview", the "Iteration" is still "Minor Revision". I believe somewhere I found the note that the handling topical editor suggested "publish as it is", now I cannot say for sure where it is. At the same time, we have not received any further comments for revision.

So we uploaded the files for the use of publication.

Best regards

Xiaoli Guo Larsén and Jana Fischereit